



# Accelerated pseudo-transient method for elastic, viscoelastic, and coupled hydro-mechanical problems with applications

Yury Alkhimenkov[1] and Yury Podladchikov[2]

[1]Department of Civil and Environmental Engineering, Massachusetts Institute of Technology, Cambridge, MA 02139, USA
[2]Institute of Earth Sciences, University of Lausanne, Switzerland

**Correspondence:** Yury Alkhimenkov (yalkhime@mit.edu)

**Abstract.** The Accelerated Pseudo-Transient (APT) method is a matrix-free approach used to solve partial differential equations (PDEs), characterized by its reliance on local operations, which makes it highly suitable for parallelization. With the advent of the memory-wall phenomenon around 2005, where memory access speed overtook floating-point operations as the bottleneck in high-performance computing, the APT method has gained prominence as a powerful tool for tackling various
PDEs in geosciences. Recent advancements have demonstrated the APT method's computational efficiency, particularly when applied to quasi-static nonlinear problems using Graphical Processing Units (GPUs). This manuscript presents a comprehensive analysis of the APT method, focusing on its application to quasi-static elastic, viscoelastic, and coupled hydro-mechanical problems, specifically those governed by quasi-static Biot's poroelastic equations, across 1D, 2D, and 3D domains. We systematically investigate the optimal numerical parameters required to achieve rapid convergence, offering valuable insights into
the method's applicability and efficiency for a range of physical models. Our findings are validated against analytical solutions, underscoring the robustness and accuracy of the APT method in both homogeneous and heterogeneous media. We explore the influence of boundary conditions, non-linearities, and coupling on the optimal convergence parameters, highlighting the method's adaptability in addressing complex and realistic scenarios. To demonstrate the flexibility of the APT method, we apply it to the nonlinear mechanical problem of strain localization using a poro-elasto-viscoplastic rheological model, achieving
extremely high resolutions - $10,000^2$ voxels in 2D and $512^3$ voxels in 3D - that, to our knowledge, have not been previously explored for such models. Our study contributes significantly to the field by providing a robust framework for the effective implementation of the APT method in solving challenging geophysical problems. Importantly, the results presented in this paper are fully reproducible, with Matlab, symbolic Maple scripts, and CUDA C codes made available in a permanent repository.

## 1 Introduction

The Accelerated Pseudo-Transient (APT) method represents a powerful tool in computational science, combining efficiency, scalability, ease of implementation, and a strong theoretical foundation rooted in wave physics. The main idea of the APT method is that instead of solving the original partial differential equation (PDE), a modified PDE with added inertial terms and attenuation is solved in iterative fashion until the inertial terms vanish. In other words, the solution of the original PDE is an attractor of the transient PDE with inertia.





The APT method is an efficient iterative approach for solving PDEs without relying on matrix storage. This method is versatile, applicable to both linear and nonlinear equations, and distinguishes itself with several key attributes. (i) APT is a matrix-free method, enabling the solution of large-scale 3D problems without the overhead of matrix storage. (ii) leveraging only local operations, APT naturally lends itself to parallelization, making it well-suited for modern computing architectures. (iii) its structure facilitates efficient implementation on Graphical Processing Units (GPUs), capitalizing on their ability to handle parallel tasks effectively. (iv), APT method aligns closely with the physics of wave phenomena, offering a robust theoretical framework for rigorous understanding and application.

One of the first iterative methods to solve PDEs was presented by Richardson (1911). Probably one of the first iterative methods that features second-order iterations and can be called as APT was proposed in 1950s by Frankel (1950); Riley (1954) for solving elliptic equations (see also Young (1972) ). The pseudo-transient method is also known as a dynamic-relaxation (DR) method that was used by Otter (1965); Otter et al. (1966). Interestingly, the APT method was also applied in other branches of science, e.g., in areas related to optimization problems (Polyak, 1964). In geosciences, the APT method was introduced as the Fast Lagrangian Analysis of Continua (FLAC) algorithm by Cundall (1976), it was applied to solve non-linear problems and instabilities (Poliakov et al., 1993, 1994). The APT method was recently applied to model large 3D geophysical problems: coupled two-phase flow physics represented by solitary porosity waves (Räss et al., 2019), reaction-driven porosity waves (Omlin et al., 2017) and thermomechanical ice deformation (Räss et al., 2020). The APT method was applied to model focused fluid flow by Wang et al. (2022). Furthermore, Wang et al. (2022) investigated the physics-based principles underlying the APT method. A compaction-driven fluid flow and plasticity within porous media were investigated numerically by Alkhimenkov et al. (2024a). A numerical approach based on GPUs to model the strain localization in 2D and 3D of a (visco)-hypoelastic-perfectly plastic medium was developed by Alkhimenkov et al. (2024b).

The efficiency of the APT method strongly depends on the choice of the numerical parameters. For simple equations, such parameters can be derived analytically. This was done for elliptic equations by analyzing a damped wave equation (DWE) (Cox and Zuazua, 1994), since the solution of elliptic equations is an attractor of DWE. In optimization problems the APT method is also known as PDE acceleration framework (Calder and Yezzi, 2019; Benyamin et al., 2020). A comprehensive study that provides the optimal values of numerical parameters of the APT method for various problems is provided by Räss et al. (2022). Such problems include diffusion–reaction equations, transient diffusion, incompressible viscous shear-driven Couette flow, incompressible viscous and visco-elastic Stokes equation. Remarkably, the APT method can be applied to other classes of problems, that are described in the present paper.

The present study provides a comprehensive study of the application of the APT method to compressible quasi-static elastic and visco-elastic equations and to coupled hydro-mechanical problems represented by the quasi-static Biot's poroelastic equations.

The novelties of this paper are summarized as follows:

1. A set of optimal parameters tailored for compressible quasi-static elastic and viscoelastic equations is presented.

2. Validation against analytical solutions is conducted to verify the accuracy of the APT solutions of quasi-static elasticity equations.





3. A new set of optimal parameters specifically designed for coupled hydro-mechanical problems, represented by the quasi-static Biot's poroelastic equations is introduced.

   4. Applications of the APT method are presented for ultra-high resolution simulations of $10,000^2$ voxels in 2D and $512^3$ voxels in 3D for poro-elastoplastic equations.

## 2   Mathematical formulation: quasi-static elasticity equations

### 2.1   General form

Consider a domain $V$ in a three-dimensional Euclidean space $E^3$ bounded by a regular surface $\partial V$. The equilibrium equation (conservation of linear momentum under the conditions of equilibrium and neglecting body forces) is (Landau and Lifshitz, 1959; Nemat-Nasser and Hori, 2013)

$$\boldsymbol{\nabla} \cdot \boldsymbol{\sigma} = 0, \tag{1}$$

where $\boldsymbol{\sigma}$ is stress tensor, $\cdot$ is the dot product, $\boldsymbol{\nabla}$ is the del operator and $\boldsymbol{\nabla}\cdot$ is the divergence operator. The del operator, $\boldsymbol{\nabla}$, is a vectorial differential operator, denoted by Li and Wang (2008); Nemat-Nasser and Hori (2013): $\boldsymbol{\nabla} \equiv \partial_i \mathbf{e}_i \equiv \partial \mathbf{e}_i / \partial x_i$, where $\mathbf{e}_i$ are the base vectors and $x_i$ are the coordinates. The stress tensor $\boldsymbol{\sigma}$ can be decomposed into pressure (minus the mean stress), $p$, and deviatoric stress tensor, $\boldsymbol{\tau}$, such that $\boldsymbol{\sigma} = -p\mathbf{I}_2 + \boldsymbol{\tau}$, where $\mathbf{I}_2$ is the second order identity tensor. In a rate formulation, the constitutive equation (the stress-rate-velocity relation) is

$$75 \quad \frac{\partial \boldsymbol{\sigma}(\mathbf{v})}{\partial t} = \mathbf{C} : \frac{\partial \boldsymbol{\varepsilon}}{\partial t}, \tag{2}$$

$$\frac{\partial \boldsymbol{\varepsilon}}{\partial t} = \frac{1}{2}\left(\boldsymbol{\nabla} \otimes \mathbf{v} + (\boldsymbol{\nabla} \otimes \mathbf{v})^{\mathrm{T}}\right), \tag{3}$$

where $\mathbf{C}$ is the 4-th rank stiffness tensor (with components $C_{ijkl}$), ":" is the double-dot product, $\otimes$ is the tensor product, the superscript "T" denotes transpose, $\partial \boldsymbol{\varepsilon}/\partial t$ is the strain-rate tensor, $\mathbf{v}$ is the velocity field. For the elasticity problems, we consider
two different tasks: (i) loading/unloading of an elastic body and (ii) calculation of effective elastic properties.

### 2.2   1D elasticity equations

For simplicity, we consider 1D elasticity equations as the following system:

$$\begin{cases} \dfrac{\partial \sigma_{xx}}{\partial t} = \left(K + \dfrac{4}{3}G\right)\dfrac{\partial v_x}{\partial x} \\[3mm] 0 = \dfrac{\partial \sigma_{xx}}{\partial x}, \end{cases} \tag{4}$$





where $\sigma_{xx}$ is the component of the stress tensor, $v_x$ is the velocity, $K$ is the bulk modulus, $G$ is the shear modulus. Note that
the system of equations (4) is a 1D version of the full system of elasticity equations (1)-(3).

## 2.3 The pseudo-transient method

The pseudo-transient(PT) method method is used to solve the system of equation (4) (Frankel, 1950; Räss et al., 2022). The pseudo-transient method is matrix-free and builds on a transient physics analogy to establish a stationary solution. The main idea is that the solution of a quasi-static equation (stationary process), usually described by an elliptic PDE, is represented by
90 an attractor of a transient process described by parabolic or hyperbolic PDEs.

### 2.3.1 The first-order PT method

Let us write the first and the simplest version of the pseudo-transient method:

$$
\begin{cases}
\dfrac{\partial \sigma_{xx}}{\partial t} = (K + \tfrac{4}{3}G)\dfrac{\partial v_x}{\partial x} \\[4mm]
0 = \dfrac{\partial \sigma_{xx}}{\partial x} - \mu\, v_x,
\end{cases}
\tag{5}
$$

where $\mu$ is an attenuation parameter. The system of equations (5) represents a diffusive-type physical behavior. The system is
95 solved once the term $\mu v_x$ converges to zero with a certain precision (e.g., $10^{-12}$). The convergence of this type of equation is $\sim n_x^2$, where $n_x$ is the number of grid cells in x-direction. Such convergence rate makes this method impractical for large 3D problems, therefore, this method is not analyzed here. An interested reader can find more details in Räss et al. (2022).

### 2.3.2 The accelerated pseudo-transient method: damping scheme 1

Now, let us consider a more advanced version of the pseudo-transient which we will call the accelerated pseudo-transient
method (APT):

$$
\begin{cases}
\dfrac{\partial \sigma_{xx}}{\partial t} = (K + \tfrac{4}{3}G)\dfrac{\partial v_x}{\partial x} \\[4mm]
\widetilde{\rho}\dfrac{\partial v_x}{\partial \widetilde{t}} = \dfrac{\partial \sigma_{xx}}{\partial x} - \mu\, v_x,
\end{cases}
\tag{6}
$$

where $\widetilde{t}$ is a "pseudo" time and $\mu$ is an attenuation parameter. The system (6) is solved once the terms $\partial v_x/\partial \widetilde{t}$ and $\mu v_x$ converge to zero with a certain precision (e.g., $10^{-12}$). The advantage of this system of equation (6) over (5) is that now the system of equation (6) describes propagating waves (i.e., hyperbolic), and, therefore, the convergence rate is $\sim n_x$ (compare to $\sim n_x^2$ in
the first-order PT method (5)) (see Räss et al. (2022) for details). This method has been successfully applied to solve coupled two-phase flow physics represented by solitary porosity waves (Räss et al., 2019).

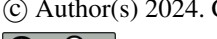



### 2.3.3 The accelerated pseudo-transient method: damping scheme 2

Here we report a modification of the APT method. The solution of the quasi-static elasticity equations can be achieved in two steps. (i) Inertial terms are added into the equations stress, (ii) these terms are treated as a Maxwell rheology (a viscous damper). The quasi-static elasticity equations (5) can then be re-written with the pseudo-time $\widetilde{t}$,

$$
\begin{cases}
\dfrac{1}{\widetilde{H}} \dfrac{\partial \sigma_{xx}}{\partial \widetilde{t}} + \dfrac{1}{H} \dfrac{\sigma_{xx} - \hat{\sigma}_{xx}}{\Delta t} = \dfrac{\partial v_x}{\partial x} \\[3mm]
\widetilde{\rho} \dfrac{\partial v_x}{\partial \widetilde{t}} = \dfrac{\partial \sigma_{xx}}{\partial x},
\end{cases}
\tag{7}
$$

where $\hat{\sigma}_{xx}$ is the stress field at the previous physical time step and $C_{1111} = \widetilde{H} \equiv H = K + \frac{4}{3}G$ is the P-wave modulus. The system (7) has to be solved for the case of elastic loading/unloading where the stress $\hat{\sigma}_{xx}$ is non-zero from the previous physical time step.

The system of equations (7) can be further simplified since the stress $\hat{\sigma}$ does not change inside the loop over "pseudo" time $\widetilde{t}$:

$$
\begin{cases}
\dfrac{1}{\widetilde{H}} \dfrac{\partial \sigma_{xx}}{\partial \widetilde{t}} + \dfrac{1}{H} \dfrac{\sigma_{xx}}{\Delta t} = \dfrac{\partial v_x}{\partial x} \\[3mm]
\widetilde{\rho} \dfrac{\partial v_x}{\partial \widetilde{t}} = \dfrac{\partial \sigma_{xx}}{\partial x}.
\end{cases}
\tag{8}
$$

In the system (7) (or (8)), $\widetilde{\rho}$ is a to be determined numerical parameter. For the analysis of the optimal numerical parameters, the systems of equations (7) and (8) are equivalent to each other since the quantity $\hat{\sigma}_{xx}$ is constant during the iterations over the "pseudo" time $\widetilde{t}$.

The APT version of expression (7) (or (8)) where the stress tenor is decomposed into pressure and deviatoric stress tensor provided in Appendix A. A discrete version of the system (8) is provided in Appendix B. A Matlab routine to solve the system (8) is presented in Appendix C.

The system of equations (8) is hyperbolic and corresponds to a wave propagation in a dissipative medium. The numerical parameters in the system (8) determine the attenuation of propagating waves. Our target is to solve elasticity equations that are quasi-static. Therefore, the goal is to find optimal values of the numerical parameters that corresponds to the fastest attenuation of propagating waves. More precisely, once the "pseudo" time derivatives ($\partial \sigma_{xx}/\partial \widetilde{t}$, $\partial v_x/\partial \widetilde{t}$) in the system (8) disappear, the resulting solution of the quasi-static equations is found. In other words, the solution to quasi-static equations in an attractor of the system of equations (8) at large "pseudo"-time-scales. For a particular (optimal) choice of the numerical parameters, the attractor solution can be achieved faster than by using non-optimal values of the numerical parameters. In the best scenario, the number of iterations $n_I$ needed to converge to the target solution is $n_I \sim n_x$, and more precisely $n_I = k\, n_x$, where usually $k$ is in a range of $k \in [5; 50]$ (the low and upper bounds provided must be considered as an approximation). In other words, the



wave travels several times throughout the whole domain before the corresponding updates of the time derivatives attenuate to a desired precision. If non-optimal parameters are used, the solution may not converge for a long computational time.

Let us describe some basic features of the system of equations (8). The "numerical" primary or P-wave velocity can be calculated as:

$$\widetilde{V}_p = \sqrt{\frac{\widetilde{H}}{\widetilde{\rho}}}. \tag{9}$$

The Courant–Friedrichs–Lewy (CFL) condition for the system of equation (8) suggest that (Alkhimenkov et al., 2021a)

$$\Delta \widetilde{t} \leq \frac{\Delta x}{\widetilde{V}_p} \qquad \text{or} \qquad \Delta \widetilde{t} = \frac{\widetilde{C} \Delta x}{\widetilde{V}_p}, \tag{10}$$

where $\widetilde{C} \leq 1$. Note that the system of equations (8) is identical to the damped linear wave equation and the CFL condition (10) is just a lower bound (Alkhimenkov et al., 2021a). It is important to mention that we do not need to know the optimal values of all the numerical parameters separately. Instead, the following combinations are needed: $\widetilde{H}\Delta\widetilde{t}$ and $\Delta\widetilde{t}/\widetilde{\rho}$.

     Let us analyze the system of equations (8). First, we perform a dispersion analysis. A solution of traveling waves in dissipative media can be written as

$$f(\widetilde{t}, x) = \exp\left[\frac{(\gamma \widetilde{V}_p \widetilde{t} + \pi \omega x i)}{L_x}\right], \tag{11}$$

where $\gamma$ is the amplitude, $\omega = 2\pi f$ is the angular frequency ($f$ is the frequency), $i$ is the imaginary unit and in our description $\exp[\cdot] \equiv e^{(\cdot)}$. The amplification matrix $F$ of this system is a $2 \times 2$ matrix:

$$F = \begin{bmatrix} \dfrac{\gamma \Delta x}{L_x} & \dfrac{-3i\pi \Delta x}{7\,\mathrm{St}} \\[3ex] \dfrac{-7\Delta x\,\mathrm{St}\,\pi}{3\,L_x^2} & \dfrac{\Delta x\,(\mathrm{St}+\gamma)}{L_x} \end{bmatrix}, \tag{12}$$

where the dimensionless parameter, the Strouhal number, $\mathrm{St}$, is expressed as

$$\mathrm{St} = \frac{L_x}{\widetilde{V}_p\,\Delta t}. \tag{13}$$

The discriminant $D$ of the matrix (12) is

$$D = \left(\gamma^2 + \mathrm{St}\,\gamma + \pi^2\right)\left(\frac{\Delta x}{L_x}\right)^2 \tag{14}$$

Setting $D = 0$ and solving for $\gamma$, we get two roots:

$$\gamma_1 = -\frac{\mathrm{St}}{2} + \frac{\sqrt{-4\pi^2 + \mathrm{St}^2}}{2}, \tag{15}$$





$$\gamma_2 = -\frac{\mathrm{St}}{2} - \frac{\sqrt{-4\pi^2 + \mathrm{St}^2}}{2}, \tag{16}$$

The minimum of real part of the roots $\gamma_1$ and $\gamma_2$ control the exponential decay rate of the solution (Räss et al., 2022), therefore, we are interested in the minimum of these values. This minimum reaches maximal value when the discriminant is zero:

$$-4\pi^2 + \mathrm{St}^2 = 0. \tag{17}$$

The resulting solution for $\mathrm{St}$ has two roots: $2\pi$ and $-2\pi$. Taking the positive root we get

$$\mathrm{St} = \mathrm{St_{opt}} = 2\pi, \tag{18}$$

which is the optimal value of the numerical parameter $\mathrm{St}$ that corresponds to the fastest attenuation of propagating waves.

There is only one numerical parameter that controls the dissipation and convergence to the target solution of the quasi-static equations: the Strouhal number, $\mathrm{St}$, which is a purely numerical parameter in our analysis and can be chosen arbitrary. For
$\mathrm{St} \ll 1$ the system of equations (8) behaves as purely hyperbolic without the stiff source term; in other words, propagating waves do not attenuate (especially when $\mathrm{St} \to 0$). Contrary, for $\mathrm{St} \gg 1$ the system of equations (8) behaves as hyperbolic with the stiff source term, that dominates; therefore, the system of equations (8) behaves as a diffusion process and attenuate very slowly. The optimal choice of the Strouhal number, $\mathrm{St}$, is between these two limits: $\mathrm{St} = \mathrm{St_{opt}} = 2\pi$ as it is shown by expression (18).

Let us do some transformations with expression (13). Our goal is to separate the numerical combination $\Delta\widetilde{t}/\widetilde{\rho}$ on the left hand side and the other variables on the right hand side:

$$1 = \frac{L_x}{\mathrm{St}\,\widetilde{V}_p\,\Delta t} \iff 1 = \frac{L_x\sqrt{\widetilde{\rho}}}{\mathrm{St}\,\sqrt{\widetilde{H}}\,\Delta t}\frac{\sqrt{\widetilde{\rho}}\Delta\widetilde{t}}{\sqrt{\widetilde{\rho}}\Delta\widetilde{t}} \iff \frac{\Delta\widetilde{t}}{\widetilde{\rho}} = \frac{L_x\Delta\widetilde{t}}{\mathrm{St}\,\sqrt{\widetilde{H}}\,\sqrt{\widetilde{\rho}}\,\Delta t}\frac{\widetilde{V}_p}{\widetilde{V}_p}, \tag{19}$$

and continue

$$\frac{\Delta\widetilde{t}}{\widetilde{\rho}} = \frac{\widetilde{V}_p\,\Delta\widetilde{t}\,L_x}{\mathrm{St}\,\widetilde{H}\,\Delta t}. \tag{20}$$

By using expression (10), we evaluate that $\widetilde{V}_p\,\Delta\widetilde{t} = \widetilde{C}\Delta x$, therefore, equation (20) can be rewritten as

$$\frac{\Delta\widetilde{t}}{\widetilde{\rho}} = \frac{\widetilde{C}\Delta x\,L_x}{\mathrm{St}\,\widetilde{H}\,\Delta t}. \tag{21}$$

In expression (21), all the parameters on the right hand side are known, thus $\Delta\widetilde{t}/\widetilde{\rho}$ can be evaluated. Now let us create an expression for the second numerical combination, $\widetilde{H}\Delta\widetilde{t}$. For that we employ the following transformations:

$$1 = \frac{\widetilde{V}_p^2\,\Delta\widetilde{t}^2}{\widetilde{V}_p^2\,\Delta\widetilde{t}^2} \iff \frac{\widetilde{V}_p^2\,\widetilde{\rho}\,\Delta\widetilde{t}^2}{\widetilde{H}\,\Delta\widetilde{t}^2} \iff \widetilde{H}\Delta\widetilde{t} = (\widetilde{V}_p\,\Delta\widetilde{t})^2\left(\frac{\Delta\widetilde{t}}{\widetilde{\rho}}\right)^{-1}, \tag{22}$$

Note that $\widetilde{V}_p\,\Delta\widetilde{t}$ and $\Delta\widetilde{t}/\widetilde{\rho}$ are already defined above, therefore, it is straightforward to calculate $\widetilde{H}\Delta\widetilde{t}$. Therefore, the system of equations (7) (or (8)) or its discrete version (B1) can be solved.





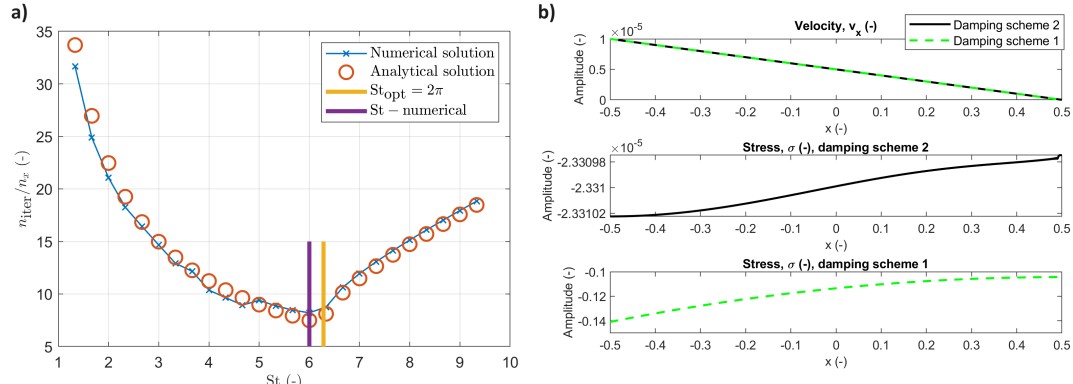

**Figure 1.** Panel (a): Convergence rate in a homogeneous elastic medium: numerical and analytical results as a function of the dimensionless parameter St. Panel (b): Numerical results for velocity and stress fields in the homogeneous medium for the two damping schemes. Upper panel corresponds to the velocity field, middle panel shows the stress field considering damping scheme 2 and lower panel shows the stress field considering damping scheme 1.

### 2.3.4 Numerical experiment 1: convergence rate in a homogeneous medium

Figure 1 shows the numerical and analytical results for the system of equations (8). The numerical results correspond to the solution with different St numbers until the update of the "pseudo-time" derivatives becomes less than $10^{-9}$. The analytical
result corresponds to the analytical solution of the dispersion relations as a function of St. It can be seen that the analytical and numerical results are in excellent agreement (Figure 1) that validates the proposed approach.

### 2.3.5 Numerical experiment 2: effective properties of a homogeneous medium

Let us consider a 1D numerical domain with $L_x = 1$, which is discretized into $n_x = 1000$ grid cells. The material parameters are $K = G = 1$ and $\Delta t = 1$. For this experiment, a velocity boundary conditions are applied by prescribing $v_x(n = 1) = 1$ and
190 $v_x(n = n_x) = 0$, where $n$ is a grid cell number in a 1D domain ($v_x(n = 1) = 1$ means that the velocity $v_x = 1$ at the first grid cell ($n = 1$) which corresponds to the left corner of the 1D domain $L_x$). All other parameters and initial conditions are set to zero.

Figure 1b shows the velocity field (panel a) and the amplitudes of the stress field for the two damping schemes (scheme 1, where $\mu = \pi$, and scheme 2). Since the medium is homogeneous, the effective elastic parameters can be calculated exactly:
$H^* = K + 4/3 G = 7/3$. Numerically, the effective elastic parameters are calculated from the discrete values for the APT method:

$$H^* = \frac{\sum_{i=1}^{nx}[\sigma_{xx}]_i}{\sum_{i=1}^{nx}[\partial u_x/\partial x]_i}, \tag{23}$$

where $u_x = v_x \Delta t$. After $5\,n_x$ iterations in "pseudo-time", both damping strategies provide us with similar results: the accuracy is $\partial v_x/\partial \widetilde{t} = 10^{-13}$ for the damping scheme 1 and $\partial v_x/\partial \widetilde{t} = 10^{-14}$ for the damping scheme 2.



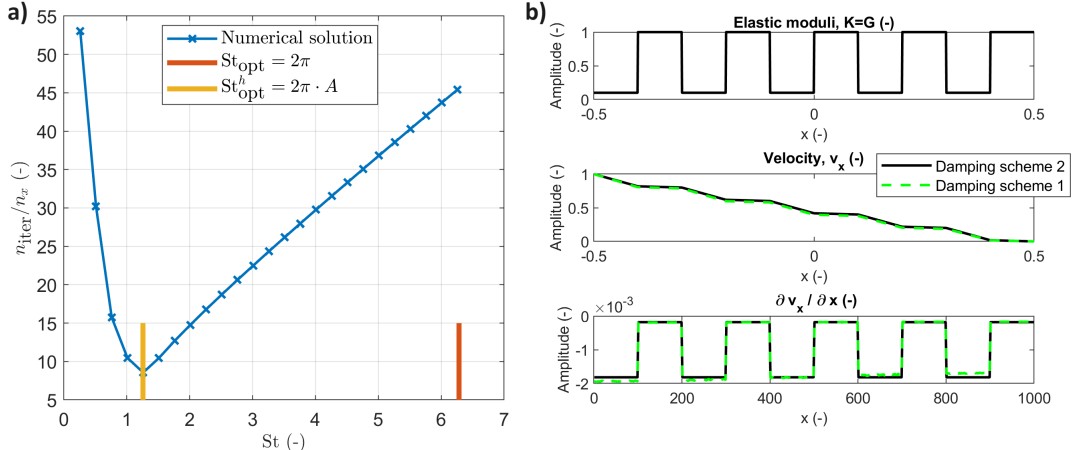

**Figure 2.** Panel (a): Numerical results: convergence rate in a heterogeneous medium for the damping scheme 2 as a function of $\mathrm{St}$. Panel (b): Numerical results for velocity and stress fields in a layered (heterogeneous) medium for the two damping schemes. Upper panel corresponds to variations of bulk modulus K (the same as variations in the shear modulus G), middle panel shows the velocity field considering damping schemes 1 and 2. Lower panel shows the spatial derivative of the velocity field considering damping schemes 1 and 2.

### 2.3.6 Numerical experiment 3: convergence rate in a heterogeneous medium

Now, let us consider a heterogeneous medium in 1D represented by layers of different elastic properties. There are ten layers with the properties $K_1 = G_1 = 1$ and $K_2 = G_2 = 0.05$. Figure 2 shows numerical results for the system of equations (8). The numerical results correspond to the solution as a function of $\mathrm{St}$ until the update of the "pseudo-time" derivatives becomes withing the range $10^{-9}$. It can be seen that the optimal value for of $\mathrm{St}$ that is valid in homogeneous medium is not valid here for a heterogeneous medium. Instead, a special scaling is needed of $\mathrm{St}$ with a parameter $A$ which is defined below.

### 2.3.7 Numerical experiment 4: effective properties of a heterogeneous medium

We perform numerical experiment considering the two damping schemes: damping schemes 1 with $\mu = \pi$ and damping schemes 2 as a function of $\mathrm{St}$. By running a set of numerical simulations with different optimal parameters, we found that the following re-scaling of $\mathrm{St}_{\mathrm{opt}}$ via parameter $A$ provides the best fast convergence rate

$$\mathrm{St}_{\mathrm{opt}}^h = A \cdot \mathrm{St}_{\mathrm{opt}}, \tag{24}$$

where $A$ is a minimum of the elastic moduli of the softest material divided by volume fraction $\phi$:

$$A = min(K_2, G_2)/\phi. \tag{25}$$

Figure Figure 2 shows the distribution of elastic moduli (panel a), the velocity field and the spatial derivative of the velocity field for damping schemes 1 and 2. After $5n_x$ iterations in "pseudo" time, damping strategies provide us with the following



results: the accuracy is $(2.5 \cdot 10^{-4})\%$ for the damping scheme 2 and $6.15\%$ for the damping scheme 1. This experiment shows that for practical applications in heterogeneous media, damping scheme 2 provides us with a faster convergence and more accurate results than the damping scheme 1. Note that the definition of $A$ in equation (25) is valid for the specific parameters of the medium considered here and is not universal.

## 3  Mathematical formulation: viscoelasticity

Now, let us consider viscoelastic equations. The system of equations (4) can be re-written for calculation of effective viscoelastic properties as

$$
\begin{cases}
\dfrac{1}{K}\dfrac{\partial p}{\partial t} = -\dfrac{\partial v_x}{\partial x} \\[2ex]
\dfrac{1}{2G}\dfrac{\partial \tau_{xx}}{\partial t} + \dfrac{\tau_{xx}}{2\mu_s} = \left( \dfrac{\partial v_x}{\partial x} - \dfrac{1}{3}\dfrac{\partial v_x}{\partial x} \right) \\[2ex]
0 = \dfrac{\partial(-p + \tau_{xx})}{\partial x},
\end{cases}
\tag{26}
$$

where $\mu_s$ is the (physical) viscosity of the solid material.

### 3.1  Naive APT scheme

The advantage of this naive APT scheme is that there are minimal modifications to the original formulation of the APT method for elasticity equations presented in the previous sections. The system of equations (26) can be re-written as the APT scheme 2:

$$
\begin{cases}
\dfrac{1}{\widetilde{K}}\dfrac{\partial p}{\partial \widetilde{t}} + \dfrac{1}{K}\dfrac{p - \hat{p}}{\Delta t} = -\dfrac{\partial v_x}{\partial x} \\[2ex]
\dfrac{1}{2\widetilde{G}}\dfrac{\partial \tau_{xx}}{\partial \widetilde{t}} + \dfrac{1}{2G}\dfrac{\tau_{xx} - \hat{\tau}_{xx}}{\Delta t} + \dfrac{\tau_{xx}}{2\mu_s} = \left( \dfrac{\partial v_x}{\partial x} - \dfrac{1}{3}\dfrac{\partial v_x}{\partial x} \right) \\[2ex]
\widetilde{\rho}\dfrac{\partial v_x}{\partial \widetilde{t}} = -\dfrac{\partial \sigma_{xx}}{\partial x},
\end{cases}
\tag{27}
$$

where

$$
\frac{\Delta \widetilde{t}}{\widetilde{\rho}} = \frac{\widetilde{V}_p \Delta \widetilde{t} L_x}{\mathrm{St}\, H^{\mathrm{ve}}},
\tag{28}
$$

and $\widetilde{H}^{\mathrm{ve}}$ is defined as

$$
H^{\mathrm{ve}} = \left( \frac{1}{(K + \frac{4}{3}G)\Delta t} + \frac{1}{\mu_s} \right)^{-1}.
\tag{29}
$$



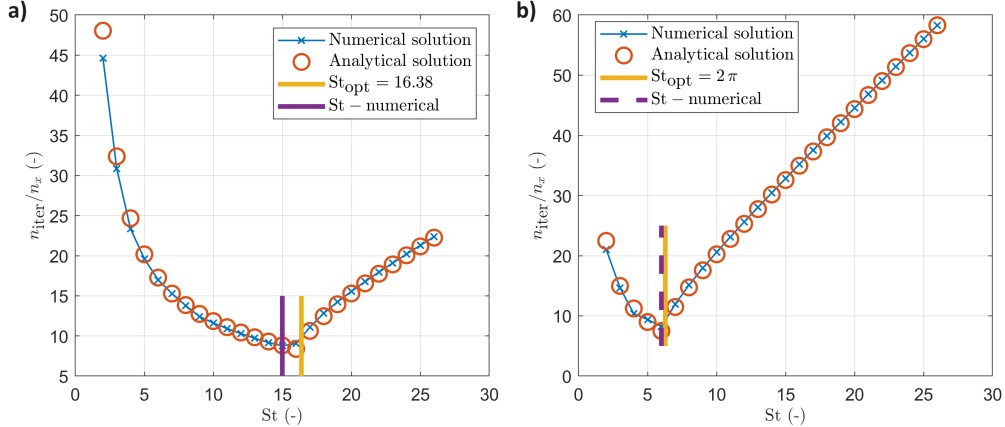

**Figure 3.** Convergence rate in a homogeneous viscoelastic medium: numerical and analytical results as a function of the dimensionless parameter St.

In other words, in the viscoelastic scenario we replaced "elastic" modulus $H$ by the viscoelastic one $\widetilde{H}^{\mathrm{ve}}$ represented by equation (29). All other numerical parameters remains the same as in single-phase elasticity equation. The resulting solution

for St has three roots. Let us assume that $K = G = \mu_s$. The positive root is

$$\mathrm{St} = \mathrm{St}_{\mathrm{opt}} = 11.469\,G + 4.915, \tag{30}$$

which is the optimal value of the numerical parameter St that corresponds to the fastest attenuation of propagating waves. If $G = 1$, then

$$\mathrm{St} = \mathrm{St}_{\mathrm{opt}} = 11.469\,G + 4.915 = 16.38. \tag{31}$$

A discrete version of the system (27) is provided in Appendix D. A similarity with the analysis proposed by Räss et al. (2022) is provided in the discussion section. Note that the optimal value of $\mathrm{St}_{\mathrm{opt}}$ must be re-evaluated for the specific parameters of the medium.

### 3.2 Elegant APT scheme

Let us simplify the scheme (27) by re-arranging terms and removing quantities that are constant during the iterations over $\widetilde{t}$:

$$\begin{cases} \dfrac{1}{\widetilde{K}}\dfrac{\partial p}{\partial \widetilde{t}} + \dfrac{1}{K}\dfrac{p}{\Delta t} = -\dfrac{\partial v_x}{\partial x} \\[2mm] \dfrac{1}{2\widetilde{G}}\dfrac{\partial \tau_{xx}}{\partial \widetilde{t}} + \dfrac{\tau_{xx}}{2}\left(\dfrac{1}{G\Delta t} + \dfrac{1}{\mu_s}\right) = \left(\dfrac{\partial v_x}{\partial x} - \dfrac{1}{3}\dfrac{\partial v_x}{\partial x}\right) \\[2mm] \widetilde{\rho}\dfrac{\partial v_x}{\partial \widetilde{t}} = -\dfrac{\partial \sigma_{xx}}{\partial x}. \end{cases} \tag{32}$$





Further simplifications leads to the following system:

$$
\begin{cases}
\dfrac{1}{\widetilde{K}}\dfrac{\partial p}{\partial \widetilde{t}} + \dfrac{1}{K}\dfrac{p}{\Delta t} = -\dfrac{\partial v_x}{\partial x} \\[2ex]
\dfrac{1}{2\widetilde{G}}\dfrac{\partial \tau_{xx}}{\partial \widetilde{t}} + \dfrac{\tau_{xx}}{2}\dfrac{1}{G^{\mathrm{ve}}} = \left(\dfrac{\partial v_x}{\partial x} - \dfrac{1}{3}\dfrac{\partial v_x}{\partial x}\right) \\[2ex]
\widetilde{\rho}\dfrac{\partial v_x}{\partial \widetilde{t}} = -\dfrac{\partial \sigma_{xx}}{\partial x},
\end{cases}
\tag{33}
$$

where

$$
G^{\mathrm{ve}} = \left(\frac{1}{G\Delta t} + \frac{1}{\mu_s}\right)^{-1}
\tag{34}
$$

is the apparent "viscoelastic" shear modulus. Note that (e.g., assuming $G^{\mathrm{ve}} = G$) the present system (33) becomes identical to the system (A4) (or (8) ) which corresponds to the elasticity equations. Therefore, all the analysis presented for elasticity equations in the previous sections can be applied to the viscoelastic equations. If $K = G^{\mathrm{ve}} = 1$, then

$$
\mathrm{St}_{\mathrm{opt}} = 2\pi,
\tag{35}
$$

which is the same value as in the case of the elasticity equations.

## 4    Mathematical formulation: coupled hydro-mechanics — quasi-static poroelasticity

The first order velocity-stress system of Biot's equations in 1D can be written as (Biot, 1962)

$$
\begin{pmatrix} \dfrac{\partial \bar{p}}{\partial t} \\[2ex] \dfrac{\partial p_f}{\partial t} \end{pmatrix} = -K_u \begin{pmatrix} 1 & B \\[1ex] B & \dfrac{B}{\alpha} \end{pmatrix} \begin{pmatrix} \dfrac{\partial v_x^s}{\partial x} \\[2ex] \dfrac{\partial q_x^D}{\partial x} \end{pmatrix},
\tag{36}
$$

$$
\frac{\partial \bar{\tau}_{xx}}{\partial t} = 2G_u \left(\frac{\partial v_x}{\partial x} - \frac{1}{3}\frac{\partial v_x}{\partial x}\right)
\tag{37}
$$

and

$$
\begin{pmatrix} 0 \\[1ex] 0 \end{pmatrix} = \begin{pmatrix} \dfrac{\partial(-p + \tau_{xx})}{\partial x} \\[2ex] \dfrac{\eta_f}{k}q_x^D + \dfrac{\partial p_f}{\partial x} \end{pmatrix},
\tag{38}
$$

The list of symbols is given in Table 1. From the general principles of thermodynamic, the matrices of coefficients in expression (36) must be positive definite. For simplicity, expressions (36) and (37) can be combined, leading to

$$
\begin{pmatrix} \dfrac{\partial \bar{\sigma}_{xx}}{\partial t} \\[2ex] -\dfrac{\partial p_f}{\partial t} \end{pmatrix} = \begin{pmatrix} K_u + \dfrac{4}{3}G_u & K_u B \\[2ex] K_u B & \dfrac{K_u B}{\alpha} \end{pmatrix} \begin{pmatrix} \dfrac{\partial v_x^s}{\partial x} \\[2ex] \dfrac{\partial q_x^D}{\partial x} \end{pmatrix},
\tag{39}
$$





**Table 1.** List of Symbols

| Symbol | Meaning |
| --- | --- |
| $\sigma^s, \sigma^f$ | solid and fluid stress |
| $\bar{\sigma}$ | $= (1-\phi)\sigma^s + \phi\sigma^f$, total stress |
| $p_s, p_f$ | solid and fluid pressure |
| $\bar{p}$ | $= (1-\phi)p_s + \phi p_f$, total pressure |
| $\bar{\tau}_{xx}$ | total deviatoric stress |
| $v^s, v^f$ | solid and fluid velocity |
| $q^D$ | $= \phi(v^f - v^s)$, Darcy velocity |
| $\rho^s, \rho^f$ | solid and fluid density |
| $\rho_t$ | $= (1-\phi)\rho_s + \phi\rho_f$, total density |
| $K_s, K_f$ | elastic solid and fluid bulk modulus |
| $G_s, G_d = G_u$ | elastic solid, drained and undrained shear modulus |
| $K_d, K_u$ | elastic drained and undrained bulk modulus |
| $\eta_f$ | fluid shear viscosity |
| $k$ | medium permeability |
| $\phi$ | medium porosity |
| $\alpha$ | Biot-Willis coefficient |
| $B$ | Skempton coefficient |



where $\bar{\sigma}_{xx} = -\bar{p} + \bar{\tau}_{xx}$. For an isotropic material saturated with a single fluid, in which the solid frame consists of a single isotropic mineral, the Biot-Willis coefficient is

$$\alpha = 1 - \frac{K_d}{K_s} \tag{40}$$

and the Skempton coefficient, $B$, is

$$B = \frac{1/K_d - 1/K_s}{1/K_d - 1/K_s + \phi(1/K_f - 1/K_s)}. \tag{41}$$

Other useful parameters include the undrained bulk modulus, $K_u$,

$$K_u = K_d (1 - \alpha B)^{-1} \equiv K_d + \alpha^2 M, \tag{42}$$

and the fluid storage coefficient, $M$,

$$M = K_u B / \alpha. \tag{43}$$

Equaition (42) is known as Gassmann's equation for fluid-saturated bulk modulus (Gassmann, 1951; Alkhimenkov, 2023).

## 4.1 APT method for the quasi-static Biot's poroelastic equations

Let us write the APT method (scheme 2) for the quasi-static Biot's poroelastic equations (36)-(38):

$$\begin{pmatrix} \dfrac{1}{\widetilde{K}_1}\dfrac{\partial \bar{p}}{\partial \widetilde{t}} \\ \dfrac{1}{\widetilde{K}_2}\dfrac{\partial p_f}{\partial \widetilde{t}} \end{pmatrix} + \frac{1}{K_u} \begin{pmatrix} \dfrac{\bar{p}-\hat{p}}{\Delta t} \\ \dfrac{p_f - \hat{p}_f}{\Delta t} \end{pmatrix} = - \begin{pmatrix} 1 & B \\ B & \alpha \end{pmatrix} \begin{pmatrix} \dfrac{\partial v^s}{\partial x} \\ \dfrac{\partial q^D}{\partial x} \end{pmatrix}, \tag{44}$$

where $\hat{\bar{p}}$ and $\hat{p}_f$ are the total and fluid pressures at the previous physical time step, $\widetilde{K}_1 = \widetilde{K}_2 = K_u$. For the total deviatoric stress the corresponding equation is

$$\frac{1}{2\widetilde{G}_1}\frac{\partial \bar{\tau}_{xx}}{\partial \widetilde{t}} + \frac{1}{2G_u}\frac{\bar{\tau}_{xx} - \hat{\bar{\tau}}_{xx}}{\Delta t} = \left( \frac{\partial v_x}{\partial x} - \frac{1}{3}\frac{\partial v_x}{\partial x} \right), \tag{45}$$

where $\hat{\bar{\tau}}_{xx}$ is the total stress deviator at the previous physical time step and $\widetilde{G}_1 = G_u$. The system of equation (38) is re-written as

$$\begin{pmatrix} \widetilde{\rho}_t & 0 \\ 0 & \widetilde{\rho}_a \end{pmatrix} \begin{pmatrix} \dfrac{\partial v_i^s}{\partial \widetilde{t}} \\ -\dfrac{\partial q_i^D}{\partial \widetilde{t}} \end{pmatrix} = \begin{pmatrix} \dfrac{\partial(-p + \tau_{xx})}{\partial x} \\ \dfrac{\eta_f}{k}q_i^D + \dfrac{\partial p_f}{\partial x} \end{pmatrix}, \tag{46}$$

where $\widetilde{\rho}_t$ and $\widetilde{\rho}_a$ are to be determined numerical parameters. A discrete form of the system of equations (44)-(46) is presented in Appendix E. In summary, we need the following combinations of the numerical parameters to effectively solve the system of equations (44)-(46): $\widetilde{K}_1\Delta\widetilde{t}$, $\widetilde{K}_2\Delta\widetilde{t}$, $\widetilde{G}_u\Delta\widetilde{t}$, $\Delta\widetilde{t}/\widetilde{\rho}_t$ and $\Delta\widetilde{t}/\widetilde{\rho}_a$. A dispersion analysis of equations (44)-(46) leads to the system



of 5 equations. Without the loss of generality, we analyze the APT method of the expressions (39) and (38) which corresponds to the system of 4 equations in the dispersion analysis.

The "numerical" primary or P-wave velocity of the system of equations (44)-(46) varies as a function of $I_2$, which is a non-dimensional parameter:

$$I_2 = \frac{\eta_f}{k} \widetilde{\rho}_a \tau^*, \tag{47}$$

where $\tau^*$ is a characteristic time. For details on the non-dimensional analysis of these equations we refer to Alkhimenkov et al. (2021b). The CFL condition for the system of equation (44)-(46) suggest that (Alkhimenkov et al., 2021a)

$$\Delta \widetilde{t} \leq \frac{\Delta x}{\widetilde{V}_p^{HF}} \qquad \text{or} \qquad \Delta \widetilde{t} = \frac{\widetilde{C} \Delta x}{\widetilde{V}_p^{HF}}, \tag{48}$$

where $\widetilde{V}_p^{HF}$ is the "numerical" P-wave velocity at high frequencies and $\widetilde{C} \leq 1$. Note that $\widetilde{V}_p^{HF} > \widetilde{V}_p^{LF}$, where the latter is the "numerical" P-wave velocity at low frequencies. Since the exact expression for $\widetilde{V}_p^{HF}$ is cumbersome, we can modify the CFL condition (48) as

$$\Delta \widetilde{t} = \frac{\widetilde{C} \Delta x}{\widetilde{V}_p^{LF}}, \tag{49}$$

where

$$\widetilde{V}_p^{LF} = \sqrt{\frac{\widetilde{K}_1 + \frac{4}{3}\widetilde{G}_1}{\widetilde{\rho}}} = \sqrt{\frac{K_u + \frac{4}{3}G_u}{\widetilde{\rho}}} = \sqrt{\frac{H_u}{\widetilde{\rho}}}, \tag{50}$$

where $H_u = K_u + \frac{4}{3}G_u$ is the undrained P-wave modulus.

### 4.1.1 The choice of the numerical parameters

The analysis here is similar to that one for a single-phase media. From the stability analysis (49), we evaluate that

$$\widetilde{V}_p^{LF} \Delta \widetilde{t} = \widetilde{C} \Delta x. \tag{51}$$

Let us introduce a dimensionless parameter, the Strouhal number (St) which is expressed as

$$\text{St} = \frac{L_x}{\widetilde{V}_p^{LF} \Delta t}. \tag{52}$$

By analogy with expression (19), we write the formula for the first numerical combination:

$$\frac{\Delta \widetilde{t}}{\widetilde{\rho}_t} = \frac{\widetilde{V}_p^{LF} \Delta \widetilde{t} L_x}{\text{St} H_u \Delta t}. \tag{53}$$

The second numerical combination is

$$\widetilde{G}_1 \Delta \widetilde{t} = \frac{(\widetilde{V}_p^{LF} \Delta \widetilde{t})^2}{(r + \frac{4}{3})} \left( \frac{\Delta \widetilde{t}}{\widetilde{\rho}_t} \right)^{-1}, \tag{54}$$





where $r = K_u/G_u$. Note that $\widetilde{V}_p^{LF}\,\Delta\widetilde{t}$ and $\Delta\widetilde{t}/\widetilde{\rho}_t$ are already defined above, therefore, it is straightforward to calculate $\widetilde{G}_1\Delta\widetilde{t}$. Calculation of $\widetilde{K}_1\Delta\widetilde{t}$ is also straightforward: $\widetilde{K}_1\,\Delta\widetilde{t} = r\,\widetilde{G}_1\,\Delta\widetilde{t}$. For the last combination $\Delta\widetilde{t}/\widetilde{\rho}_a$, we explore the discrete system of equations and find that

$$\frac{\Delta\widetilde{t}}{\widetilde{\rho}_a} = \frac{\Delta\widetilde{t}}{\widetilde{\rho}_t}\frac{\eta_f}{k}. \tag{55}$$

Now, the system of equations (44)-(46) can be solved.

In order to find the optimal values of $\mathrm{St}$, we perform the same dispersion analysis as for a single phase media. A solution of traveling waves in dissipative media is

$$f(\widetilde{t},x) = \exp\left[\frac{(\gamma\,\widetilde{V}_p^{LF}\,\widetilde{t} + \pi\,\omega\,x\,i)}{L_x}\right], \tag{56}$$

where $\gamma$ is the amplitude, $\omega = 2\pi\,f$ is the angular frequency ($f$ is the frequency), $i$ is the imaginary unit and in our descrip-
tion $\exp[\cdot] \equiv e^{(\cdot)}$. A dispersion analysis of the system of equations (39) and (38) leads to a $4\times4$ amplification matrix. The discriminant of this matrix has four roots. The optimal value of $\mathrm{St}$ that corresponds to the fastest attenuation of propagating waves depends on the parameter $I_2$. Let us consider two end-member scenarios for values of $I_2$, which are explored in the next section.

### 4.1.2   APT method for $I_2 \ggg 1$

Figure 4a shows the numerical and analytical results for the system of equations (44)-(46) for $I_2 = 1000$. The numerical results correspond to the solution with different $\mathrm{St}$ numbers until the update of the "pseudo-time" derivatives becomes less than $10^{-11}$. The analytical result corresponds to the analytical solution of the dispersion relations as a function of $\mathrm{St}$. It can be seen that the analytical and numerical results are in excellent agreement (Figure 4a) that validates the proposed approach. Here $\mathrm{St}_{\mathrm{opt}}$ is

$$\mathrm{St} = \mathrm{St}_{\mathrm{opt}} \approx 2\,\pi. \tag{57}$$

### 4.1.3   APT method for $I_2 \lll 1$

Figure 4b shows the numerical and analytical results for the system of equations (44)-(46) for $I_2 = 0.001$. The numerical results correspond to the solution with different $\mathrm{St}$ numbers until the update of the "pseudo-time" derivatives becomes less than $10^{-11}$. The analytical result corresponds to the analytical solution of the dispersion relations as a function of $\mathrm{St}$. It can be seen that the analytical and numerical results are in a good agreement (Figure 4b) that validates the proposed approach. Here
$\mathrm{St}_{\mathrm{opt}}$ is

$$\mathrm{St} = \mathrm{St}_{\mathrm{opt}} \approx 2.9. \tag{58}$$

Figure 5 shows the analytical results for the system of equations (44)-(46) as a function of the dimensionless parameter $\mathrm{St}$ and $I_2$ (by varying $\eta_f/k$ only). Note that the optimal value of $\mathrm{St}$ depends on the value $I_2$.





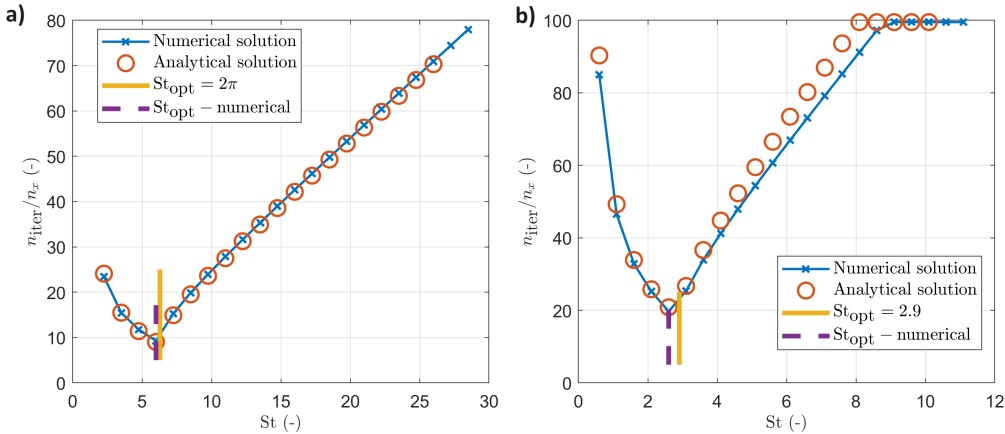

**Figure 4.** Panel (a): Convergence rate in a homogeneous poroelastic medium for $I_2 = 1000$: numerical and analytical results as a function of the dimensionless parameter St. Panel (b): Convergence rate in a homogeneous poroelastic medium for $I_2 = 0.001$: numerical and analytical results as a function of the dimensionless parameter St.

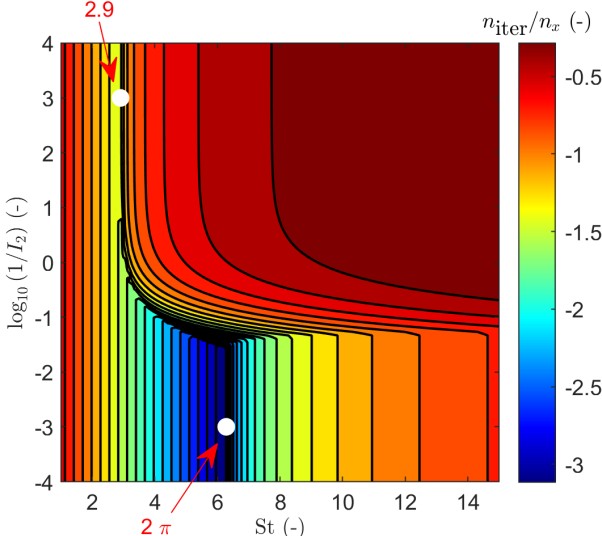

**Figure 5.** Convergence rate in a homogeneous poroelastic medium as a two-dimensional plot: analytical results as a function of the dimensionless parameter St and $I_2$. The two white circles correspond to the values of St obtained via expressions (57) and (58).





### 4.1.4 Approximation: reduced order equations

To find the optimal values of optimal parameters for the system of equations (44)-(46) a solution of 5-th order (or 4-th order) polynomial is required. However, if we neglect the coupling in the stress-strain relation, we arrive to a 4-th order (consider only the 4-th order polynomial for simplicity) polynomial where the roots can be easily separated: two roots are the same as for single-phase elastic media and the other two roots are more complicated and belong to Darcy's law.

The discriminant $D$ of the matrix amplification matrix that corresponds to expressions (39) and (38) is

$$D = \frac{3}{14StI_2 + 6}[((14StI_2)/3 + 2)\gamma^2 + (14/3St^2I_2 + 2St + 2)\gamma + St(\pi^2I_2 + 2)]\cdot$$

$$\cdot(\pi^2 + St\gamma + \gamma^2). \tag{59}$$

Setting $D = 0$ and solving for $\gamma$, we get 4 roots. Two of them correspond to the term $(\pi^2 + St\gamma + \gamma^2)$ and are the same as for single-phase media:

$$\gamma_1 = -\frac{\text{St}}{2} + \frac{\sqrt{-4\pi^2 + \text{St}^2}}{2}, \tag{60}$$

$$\gamma_2 = -\frac{\text{St}}{2} - \frac{\sqrt{-4\pi^2 + \text{St}^2}}{2}, \tag{61}$$

We are interested when the discriminant is zero: $-4\pi^2 + \text{St}^2 = 0$. The resulting solution for $\text{St}$ has two roots: $2\pi$ and $-2\pi$. Taking the positive root we get

$$\text{St} = \text{St}_{\text{opt}} = 2\pi, \tag{62}$$

which is the optimal value of the numerical parameter $\text{St}$ that corresponds to the fastest attenuation of propagating waves for
$I_2 \ggg 1$.

The two other roots are are cumbersome. However, a precise analytical evaluation is possible for any value of $I_2$. The optimal parameters for the system of equations (44)-(46) for $I_2 \lll 1$ is different from $2\pi$ as can be seen in a 2D plot (Figure 5):

$$\text{St} = \text{St}_{\text{opt}} \approx 2.9. \tag{63}$$

In summary, for practical purposes there is not need to always solve a 4-th (or 5-th) order polynomial for each set of input
parameters of the quasi-static Biot's poroelastic equations. In some cases, an average of two parameters can be taken

$$\text{St} = \text{St}_{\text{opt}} \approx (2\pi + 2.9)/2 \approx 4.596. \tag{64}$$

### 4.1.5 2D and 3D numerical simulations

The accuracy of the proposed $\text{St}_{\text{opt}} \approx 2.9$ is illustrated numerically in 2D (Figure 6a-b) and in 3D (Figure 6c-d). It can be seen that the results presented here for 1D need some calibration to be applied to 2D and 3D simulations. Note that the numerical
parameters are sensitive to boundary and initial conditions. Therefore, some test must be performed for each numerical setup.



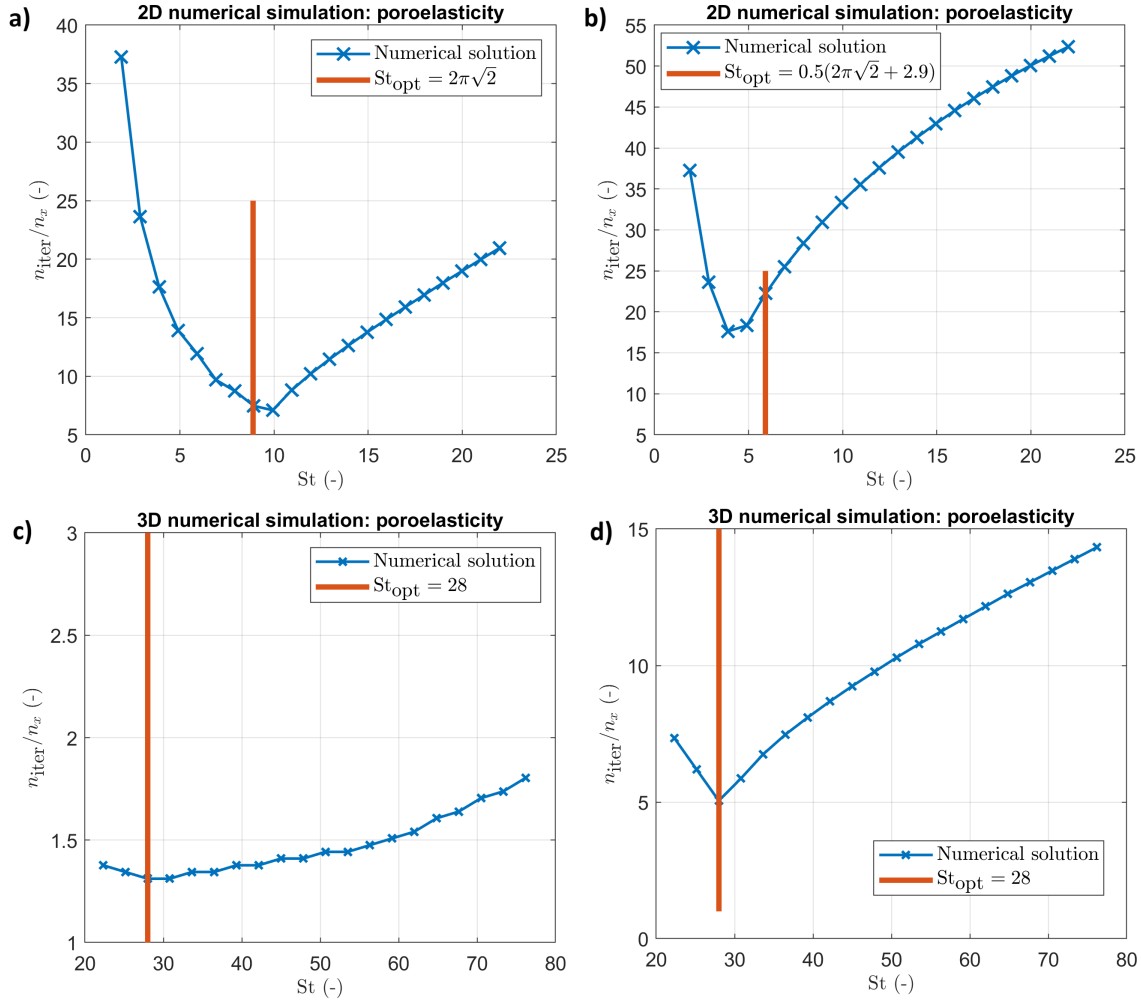

**Figure 6.** Panel (a): Convergence rate in a homogeneous poroelastic medium for $I_2 = 100$: numerical result as a function of the dimensionless parameter St. Panel (b): Convergence rate in a homogeneous poroelastic medium for $I_2 = 0.01$: numerical result as a function of the dimensionless parameter St. Panel (c): Convergence rate in a homogeneous poroelastic medium for $I_2 = 100$: numerical result as a function of the dimensionless parameter St. Panel (d): Convergence rate in a homogeneous poroelastic medium for $I_2 = 0.01$: numerical result as a function of the dimensionless parameter St.

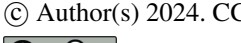



## 5 Applications: Strain Localization in Poro-Elastoplastic Media

The purpose of this section is to demonstrate the applicability of the APT method for ultra-high resolution simulations with heterogeneous initial conditions. We address the nonlinear mechanical problem of strain localization in both 2D and 3D contexts, employing an elasto-viscoplastic rheological model. This model is grounded in a hypoelastic-based constitutive framework
that accommodates the simulation of large strains. The modeling process follows the formulation of incremental constitutive equations, ensuring the objectivity of the rate fields. In this study, we utilize the Jaumann-Zaremba rate to manage the time-dependent fields.

### 5.1 Plasticity Implementation

The plasticity model adheres to a consistent poro-elasto-viscoplastic framework, with the yield function defined as

$$375 \quad F(\tau, p_e) = \sqrt{J_2} - Ap_e - Bc - \eta^{\mathrm{vp}}\dot{\lambda}, \tag{65}$$

where $\eta^{\mathrm{vp}}$ represents the viscosity of the damper, $p_e = \bar{p} - p_f$ is the effective pressure. The yield function specified by equation (65) is rate-dependent Duretz et al. (2019). The plastic potential $Q$ is expressed as

$$Q(\tau, p_e) = \sqrt{J_2} - Cp_e. \tag{66}$$

Here, the constants $A$, $B$, and $C$ are defined as $A = \sin(\phi)$, $B = \cos(\phi)$, and $C = \sin(\psi)$, where $\phi$ denotes the internal friction
angle, and $\psi \leq \phi$ is the dilation angle (with $\psi = 0$ for simplicity in this case).

In the numerical solver, plasticity is implemented through the following steps: (1) Compute the components of the trial deviatoric stresses $\bar{\tau}_{ij}^{\mathrm{trial}}$. (2) Using these components, calculate the trial second invariant of the deviatoric stresses, $J_2^{\mathrm{trial}}$. (3) Evaluate $F^{\mathrm{trial}}$ using the expression

$$F^{\mathrm{trial}} = \sqrt{J_2^{\mathrm{trial}}} - (Ap_e + Bc). \tag{67}$$

When the material remains in the plastic regime, the components of the trial deviatoric stresses, $\bar{\tau}_{ij}^{\mathrm{trial}}$, are re-scaled according to

$$\bar{\tau}_{ij}^{\mathrm{new}} = \bar{\tau}_{ij}^{\mathrm{trial}} \left( 1 - \frac{F^{\mathrm{trial}}\Delta t G_u}{\sqrt{J_2}(\Delta t G_u + \eta^{\mathrm{vp}})} \right), \tag{68}$$

This re-scaling procedure occurs within the pseudo-transient iteration loop, and the process repeats until the components of the updated trial deviatoric stresses, $\bar{\tau}_{ij}^{\mathrm{new}}$, satisfy the condition $F^{\mathrm{trial}} = 0$, and no further re-scaling is needed. This approach
is equivalent to the standard formulation involving the plastic multiplier.

### 5.2 2D Results: Ultra-High Resolution Simulations

In this set of simulations, pure shear kinematics are imposed at the boundaries of the domain, corresponding to compression along the x-axis and extension along the y-axis. The model is initialized with pre-stresses of $\bar{\tau}_{xx} = 0.0180$, $\bar{\tau}_{yy} = -0.0180$, and



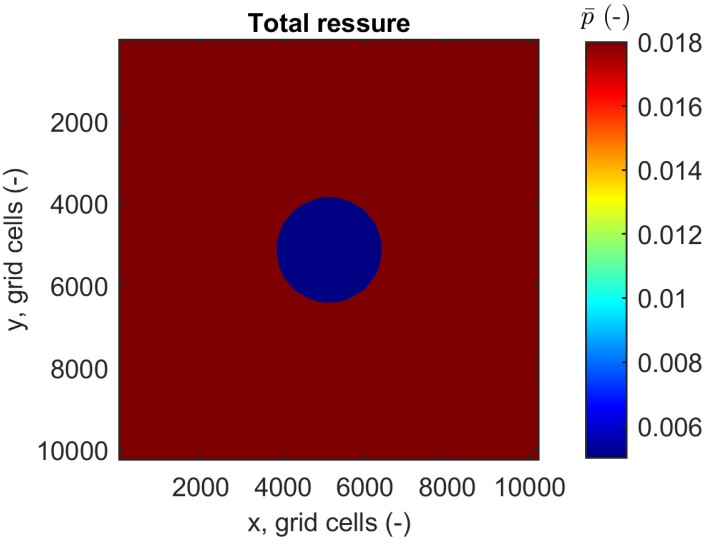

**Figure 7.** Geometry of the 2D simulation domain: a circular pressure anomaly is located at the center of the model. The resolution is $N = 10,239^2$ grid cells.

$\bar{\tau}_{xy} = 0$, while the fluid pressure $p_f$ is set to zero, and the cohesion $c$ is defined as $0.0101$. The total pressure in the background
material is $\bar{p} = 0.018$, with a circular anomaly located at the center of the model where the pressure is reduced to $\bar{p} = 0.005$
(Figure 7). The radius of this anomaly is $1/8$ of the domain size. The simulation is performed over 14 loading increments.
The poroelastic properties of the background material are: $\alpha = 0.2958$, $B = 0.0833$, $G_d = 1$, $K_d = 1$, and $\eta_f/k = 10^{-2}$. The
porosity, or fluid volume fraction, is $\phi = 0.3$, and the internal friction angle is $\varphi = 30°$.

Figure 8 shows the results of the 2D simulation with an ultra-high resolution of $N = 10,239^2$ grid cells. The finite thickness
of the shear bands confirms that the simulation is mesh-independent. The zoomed-in panels reveal extremely detailed features
of the strain localization pattern.

### 5.3 3D Results: Ultra-High Resolution Simulations

We present 3D results showcasing the spontaneous formation of shear bands under pure shear deformation, initiated by a
spherical pressure anomaly (Figure 9). These 3D simulations further validate the versatility of the APT approach (Figure 10),
demonstrating its robustness in predicting poro-elastoplastic deformation and capturing brittle failure.



**Figure 8.** 2D simulation results: snapshots of total pressure. Panel (a) shows the full model, while panels (b) and (c) present zoomed-in views of the full model. The resolution is $N = 10,239^2$ grid cells.



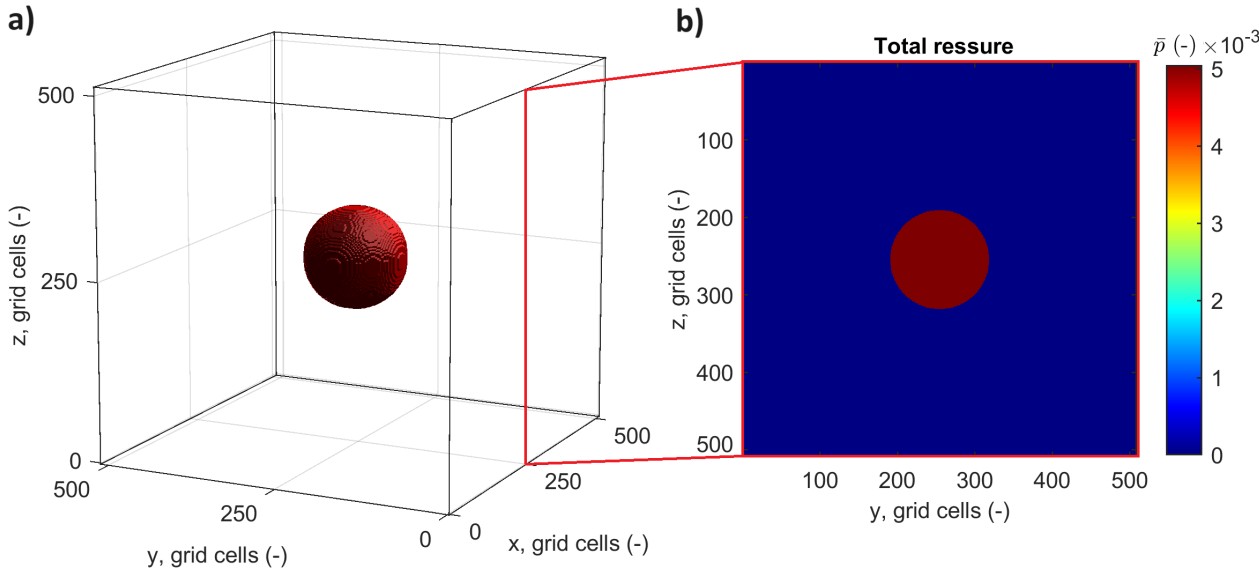

**Figure 9.** Geometry of the 3D simulation domain: a spherical pressure anomaly is located at the center. Panel (a) corresponds to 3D view, panel (b) corresponds to a slice in YZ-plane). The resolution is $N = 512^3$ grid cells.

The boundary conditions are defined by compression along the x-axis, a slight (1%) compression along the y-axis, and extension along the z-axis. The model is initialized with pre-stresses of $\bar{\tau}_{xx} = -0.0098$, $\bar{\tau}_{yy} = -9.8 \times 10^{-05}$, and $\bar{\tau}_{zz} = 0.0098$, while the shear stress components $\bar{\tau}_{xy}$, $\bar{\tau}_{xz}$, and $\bar{\tau}_{yz}$ are set to zero. The fluid pressure $p_f$ is zero, cohesion $c$ is 0.0101, and the ratio $\eta_f/k$ is set at 100. The total pressure in the background material is $\bar{p} = 0$, with a spherical anomaly located at the center of the model where the pressure is increased to $\bar{p} = 0.005$. The radius of this anomaly is $1/8$ of the domain size. The poroelastic properties of the background material are: $\alpha = 0.2958$, $B = 0.0833$, $G_d = 1$, $K_d = 1$, and $\eta_f/k = 10^2$. The porosity is $\phi = 0.3$, the internal friction angle is $\varphi = 30°$. The simulation is conducted over 15 loading increments.



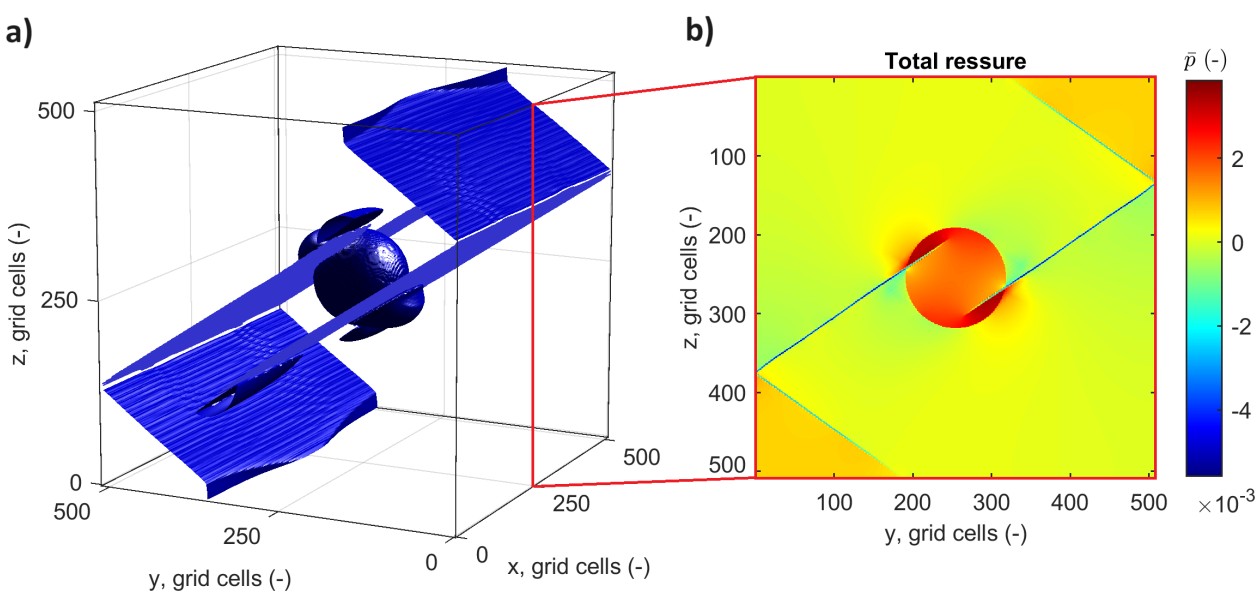

**Figure 10.** 3D simulation results: snapshots of total pressure. Panel (a) shows the 3D view of total pressure. Panel (b) shows the YZ-slice of the full 3D model.



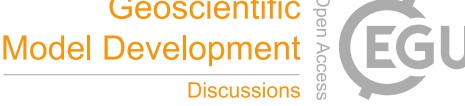

## 6 Discussion

### 6.1 Incompressible equations: a connection with the work by Räss et al. (2022)

Räss et al. (2022) performed a comprehensive analysis of the APT method for various problems. However, the work by Räss et al. (2022) was restricted mainly to single-phase media and to incompressible equations. We here provide connections of the present work to the analysis presented by Räss et al. (2022).

In the present paper we deal with compressible elastic, viscoelastic or poroelastic equations. As a result, the only numerical parameter that has to be identified is the Strouhal number, St, which is expressed as

$$
\text{St} = \frac{f L_x}{\widetilde{V}_p} = \frac{L_x}{\widetilde{V}_p \, \Delta t}, \tag{69}
$$

where $f$ is the frequency. However, in the incompressible scenario ($K \to +\infty$), an additional numerical parameter shows up: $r = \widetilde{K}/\widetilde{G}$ (which is the compressible case is defined as $r = \widetilde{K}/\widetilde{G} \equiv K/G$). Räss et al. (2022) discovered that for some specfic tasks, the value of $r$ should also be explored as well as the optimal value of St (or, equivalently, Re in their notation). As a result, Räss et al. (2022) reported the optimal values of pairs — $r$ and Re for each set of equations. A connection between the

425 "numerical" Reynolds number Re (Räss et al., 2022) and the Strouhal number St is provided below.

For the incompressible viscous Stokes equation, Räss et al. (2022) defines the "numerical" Reynolds number, Re, as

$$
\text{Re} = \frac{\widetilde{\rho} V_p^{\text{S}} L_x}{\mu_s}, \tag{70}
$$

where $V_p^{\text{S}}$ is the characteristic velocity scale for the incompressible Stokes equations

$$
V_p^{\text{S}} = \sqrt{\frac{\widetilde{K} + 2\widetilde{G}}{\widetilde{\rho}}}. \tag{71}
$$

and $\mu_s$ is the shear viscosity. Quantities $\widetilde{K}$, $\widetilde{G}$ and $\widetilde{\rho}$ are the numerical parameters. Note that in the case of incompressible viscoelastic Stokes equations, the quantity $\mu_s$ is replaced by $\mu^{\text{ve}}$:

$$
\mu^{\text{ve}} = \left( \frac{1}{G \, \Delta t} + \frac{1}{\mu_s} \right)^{-1}. \tag{72}
$$

As a result, for the incompressible viscoelastic Stokes equations, the "numerical" Reynolds number, Re, is defined as

$$
\text{Re} = \frac{\widetilde{\rho} V_p^{\text{S}} L_x}{\mu^{\text{ve}}}, \tag{73}
$$

In the present paper, from the equation (28) for viscoelastic media, we can infer the Strouhal number:

$$
\text{St} = \frac{\widetilde{\rho} \, \widetilde{V}_p \, L_x}{\widetilde{H}^{\text{ve}}} \tag{74}
$$

and $\widetilde{H}^{ve}$ is defined as

$$
H^{\text{ve}} = \left( \frac{1}{H \, \Delta t} + \frac{1}{\mu_s} \right). \tag{75}
$$





Note, that the full similarity between the definitions of $\mathrm{Re}$ (equation (73)) and Strouhal number (equation (74)). Indeed,
$\mu^{\mathrm{ve}} \equiv H^{\mathrm{ve}}$ if we neglect the physical bulk modulus $K$ (we keep only the shear modulus $G$), $V_p^{\mathrm{S}}$ is the characteristic "numerical"
velocity which has the same meaning as $\widetilde{V}_p$ for a specific problem. Therefore, all the results presented by Räss et al. (2022) for
incompressible equations can be extrapolated for compressible ones by using the results of the present paper.

### 6.2    Two- and three-dimensional simulations

As can be seen form the present study, the optimal values are similar for elastic, viscoelastic and poroelastic problems but
depend on some physical input parameters. We here report the optimal values for $\mathrm{St}$ considering elasticity equations. The
results can also be applied to viscoelastic and poroelastic problems by modifying the expressions for $\mathrm{St_{opt}}$.

Numerical tests considering elasticity equations show that the provided values for $\mathrm{St_{opt}}$ remain valid in 1D, 2D, and 3D.
However, in 2D,

$$\mathrm{St} = \mathrm{St}_{\mathrm{opt}}^{\mathrm{2D}} \approx 2\pi\sqrt{2}, \tag{76}$$

and, in 3D,

$$\mathrm{St} = \mathrm{St}_{\mathrm{opt}}^{\mathrm{3D}} \approx 2\pi\sqrt{3}. \tag{77}$$

Note that in 3D, the value of $\mathrm{St}_{\mathrm{opt}}^{\mathrm{3D}}$ can be higher and depends on the initial and boundary conditions, the medium's hetero-
geneities, and the physics involved. A typical number of iterations over the pseudo-time depends on the problem size (in grid
cells), the convergence rate and the desired precision. Form our experiments, a typical 3D heterogeneous model requires from
$5 \times n_x$ to $20 \times n_x$ ($n_x$ is the number of grid cells in x-dimension) iterations over the pseudo-time to achieve the quasi-static
solution.

### 6.3    The Influence of Boundary Conditions

Figure 11 presents 2D and 3D numerical results for the elasto-plastic medium (for the formulation, see Alkhimenkov et al.
(2024b)). The numerical outcomes are analyzed as a function of the stability parameter $\mathrm{St}$.
In the 2D simulation (Figure 11a), the total number of iterations over the pseudo-time is 3000, with a grid resolution of
$N = 511^2$ cells. The results indicate that the optimal value of $\mathrm{St}$ is $\mathrm{St} = 2\pi\sqrt{2}$, which is typically valid for homogeneous
media and appears to be approximately valid here as well, despite the slight heterogeneity of the elasto-plastic medium. This
suggests that the presence of plasticity, which introduces significant non-linearity, does not notably affect the choice of optimal
convergence parameters in this specific 2D case.
In the 3D simulations (Figures 11b-c), the total number of iterations over the pseudo-time is 1500, with a grid resolution of
$N = 191^3$ cells. For the simulation depicted in Figure 11b, pure shear boundary conditions are applied along the x- and y-axes.
In contrast, the simulation in Figure 11c uses the same model parameters but with slightly modified boundary conditions: 100%
extension along the x-axis and 50% compression along the y- and z-axes. The results reveal that the optimal value of $\mathrm{St}_{\mathrm{opt}}$ is



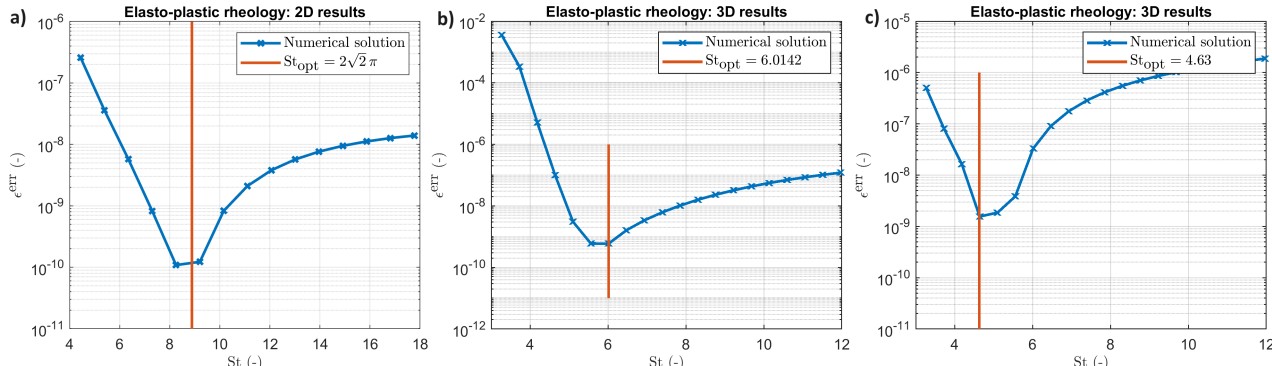

**Figure 11.** Numerical results in 2D (panel a) and 3D (panels b and c): convergence in a heterogeneous elasto-plastic medium as a function of St. The parameter $\epsilon^{\text{err}}$ corresponds to the error magnitude of the APT scheme. Simulations in panels (b) and (c) are identical except for the different partitioning of pure shear boundary conditions.

highly sensitive to the boundary conditions used. The optimal values of $\text{St}_{\text{opt}}$ discussed in the previous section are accurate
only for homogeneous media and specific initial and boundary conditions.

## 6.4   General Applicability: Influence of Initial and Boundary Conditions, Non-linearities, and Coupling

This study demonstrates that in homogeneous media with specific initial and boundary conditions, the optimal values of key numerical parameters, such as $\text{St}_{\text{opt}}$, can be accurately predicted across 1D, 2D, and 3D domains. This accuracy holds particularly true in the context of coupled systems of equations, as exemplified by the poroelastic models presented here. However,
when dealing with more complex and realistic scenarios, special considerations are required to maintain this accuracy.

Our numerical experiments highlight that factors such as initial and boundary conditions, medium heterogeneities, and the presence of coupling and non-linearities (e.g., plasticity) can significantly influence the optimal values of numerical parameters. For instance, while in homogeneous and idealized conditions, the choice of $\text{St}_{\text{opt}}$ may remain relatively stable, introducing heterogeneities or non-linear behavior, such as plasticity, necessitates a reassessment of these parameters. The study of strain
localization in both 2D and 3D models has shown that the presence of plasticity, which introduces strong non-linearities, can alter the convergence characteristics, although in some cases, the optimal parameters remain surprisingly robust.

Furthermore, the sensitivity of $\text{St}_{\text{opt}}$ to boundary conditions was particularly evident in the 3D simulations, where even minor adjustments to the boundary conditions led to changes in the optimal parameter values. This suggests that while our approach can provide a strong starting point for selecting numerical parameters, the specific conditions of each problem must
be carefully considered. In practical applications, where media may be heterogeneous, and boundary conditions complex, this study provides a framework for estimating $\text{St}_{opt}$ and other numerical parameters. However, to ensure the accuracy and efficiency of simulations, it is recommended to conduct additional test simulations. These tests are necessary to fine-tune the





parameters based on the specific characteristics of the model, such as the degree of heterogeneity, the type of non-linearities involved, and the nature of the coupling between different physical processes.

## 7 Conclusions

In this study, we conducted a rigorous analysis of the Accelerated Pseudo-Transient (APT) method for solving elastic, viscoelastic, and coupled hydro-mechanical problems, particularly those governed by Biot's poroelastic equations across 1D, 2D, and 3D domains. We identified and reported the optimal numerical parameters required to achieve rapid convergence for elastic, viscoelastic, and poroelastic problems. By systematically exploring these parameters across different spatial dimensions, we provided valuable insights into the applicability and efficiency of the APT method for a wide range of physical models.

Our study highlighted the effectiveness of the APT method in handling complex coupled systems and demonstrated its robustness across various types of media, including both homogeneous and heterogeneous conditions. By comparing our numerical results against analytical solutions for elastic equations, we validated the accuracy and reliability of the APT method in both homogeneous and heterogeneous settings.

We investigated the influence of initial and boundary conditions, non-linearities, and coupling on the optimal numerical parameters, emphasizing the importance of adaptability in real-world applications. Our findings suggest that while the APT method offers a robust framework for selecting numerical parameters, additional refinement is often necessary when dealing with heterogeneous media and complex boundary conditions. This adaptability is crucial for extending the applicability of the APT method to more realistic and challenging scenarios encountered in geomechanics and other fields involving coupled hydro-mechanical processes.

To illustrate the flexibility of the APT method, we addressed the nonlinear mechanical problem of strain localization in both 2D and 3D contexts using a poro-elasto-viscoplastic rheological model. We employed extremely high resolutions - $10,000^2$ voxels in 2D and $512^3$ voxels in 3D - which, to the best of our knowledge, have not been explored before for poro-elasto-viscoplastic rheology. This model is grounded in a hypoelastic-based constitutive framework that accommodates the simulation of large strains.

Importantly, the results presented in this paper are fully reproducible. To facilitate further research and verification, we have made available Matlab, symbolic Maple scripts, and CUDA C codes in a permanent repository.

*Code availability.* The software developed and used in the scope of this study is licensed under MIT License. The latest versions of the code is available from a permanent DOI repository (Zenodo) at: https://doi.org/10.5281/zenodo.13553494 (last access: 30 August 2024) (Alkhimenkov and Podladchikov, 2024). The repository contains code examples and can be readily used to reproduce the figures of the paper. The codes are written using the Matlab, Maple and CUDA C programming languages. Refer to the repositories' README for additional information





## Appendix A: Quasi-static elasticity equations

Let us decompose the stress tenor into pressure and deviatoric stress tensor:

$$\sigma_{xx} = -p + \tau_{xx}. \tag{A1}$$

Now, the system of equations (4) can be rewritten as

$$
\begin{cases}
\dfrac{\partial p}{\partial t} = -K\dfrac{\partial v_x}{\partial x} \\[2ex]
\dfrac{\partial \tau_{xx}}{\partial t} = 2G\left(\dfrac{\partial v_x}{\partial x} - \dfrac{1}{3}\dfrac{\partial v_x}{\partial x}\right) \\[2ex]
0 = \dfrac{\partial(-p + \tau_{xx})}{\partial x}.
\end{cases}
\tag{A2}
$$

The quasi-static elasticity equations (A2) can then be re-written with the pseudo-time $\widetilde{t}$,

$$
\begin{cases}
\dfrac{1}{\widetilde{K}}\dfrac{\partial p}{\partial \widetilde{t}} + \dfrac{1}{K}\dfrac{p - \hat{p}}{\Delta t} = -\dfrac{\partial v_x}{\partial x} \\[2ex]
\dfrac{1}{2\widetilde{G}}\dfrac{\partial \tau_{xx}}{\partial \widetilde{t}} + \dfrac{1}{2G}\dfrac{\tau_{xx} - \hat{\tau}_{xx}}{\Delta t} = \left(\dfrac{\partial v_x}{\partial x} - \dfrac{1}{3}\dfrac{\partial v_x}{\partial x}\right) \\[2ex]
\widetilde{\rho}\dfrac{\partial v_x}{\partial \widetilde{t}} = -\dfrac{\partial \sigma_{xx}}{\partial x},
\end{cases}
\tag{A3}
$$

where $\hat{p}$ is the pressure field at the previous physical time step and and $\hat{\tau}_{xx}$ in the deviatoric stress at the previous physical time step.

The system of equations (A3) can be simplified:

$$
\begin{cases}
\dfrac{1}{\widetilde{K}}\dfrac{\partial p}{\partial \widetilde{t}} + \dfrac{1}{K}\dfrac{p}{\Delta t} = -\dfrac{\partial v_x}{\partial x} \\[2ex]
\dfrac{1}{2\widetilde{G}}\dfrac{\partial \tau_{xx}}{\partial \widetilde{t}} + \dfrac{1}{2G}\dfrac{\tau_{xx}}{\Delta t} = \left(\dfrac{\partial v_x}{\partial x} - \dfrac{1}{3}\dfrac{\partial v_x}{\partial x}\right) \\[2ex]
\widetilde{\rho}\dfrac{\partial v_x}{\partial \widetilde{t}} = -\dfrac{\partial(-p + \tau_{xx})}{\partial x}.
\end{cases}
\tag{A4}
$$

$\widetilde{H} = \widetilde{K} + \frac{4}{3}\widetilde{G} = K + \frac{4}{3}G$, $\widetilde{K} = K$ and $\widetilde{G} = G$. The optimal value of St is the same as for the system (7) (or (8)):

$$\mathrm{St} = \mathrm{St}_{\mathrm{opt}} = 2\pi, \tag{A5}$$





which corresponds to the fastest attenuation of propagating waves. The stress tenor in decomposed into pressure and deviatoric stress tensor, therefore, the following expressions are also provided

$$\widetilde{G}\Delta\widetilde{t} = (\widetilde{V}_p\,\Delta\widetilde{t})^2 \left(\frac{\Delta\widetilde{t}}{\widetilde{\rho}}\right)^{-1}\left(K_G + \frac{4}{3}\right)^{-1},$$ (A6)

where $K_G = K/G$, and

$$\widetilde{K}\Delta\widetilde{t} = K_G\,\widetilde{G}\Delta\widetilde{t}.$$ (A7)

**Appendix B: Discretization: quasi-static elasticity equations**

Let us write the discrete form of the system (8). We use a classical conservative staggered space-time grid discretization (Virieux, 1986) which is equivalent to a finite volume approach (Dormy and Tarantola, 1995). More details on the present discretization can be found in Alkhimenkov et al. (2021b, a). Let us consider a physical domain $L_x$ that is discretized into grid cells such that $L_x = n_x\Delta x$. The physical time $t$ is also discretized as $\Delta t$ ($\Delta\widetilde{t}$ is the pseudo-time). The resulting discrete form of the system (8) is

$$\begin{cases} \dfrac{1}{\widetilde{H}}\dfrac{[\sigma_{xx}]_i^{l+1/2} - [\sigma_{xx}]_i^{l-1/2}}{\Delta\widetilde{t}} + \dfrac{1}{H}\dfrac{[\sigma_{xx}]_i^{l+1/2}}{\Delta t} = \dfrac{[v_x]_{i+1/2}^l - [v_x]_{i-1/2}^l}{\Delta x} \\[4mm] \widetilde{\rho}\dfrac{[v_x]_{i+1/2}^{l+1} - [v_x]_{i+1/2}^l}{\Delta\widetilde{t}} = \dfrac{[\sigma_{xx}]_{i+1}^{l+1/2} - [\sigma_{xx}]_i^{l+1/2}}{\Delta x}. \end{cases}$$ (B1)

The discrete form of the system (A3) can be written as

$$\begin{cases} \dfrac{1}{\widetilde{K}}\dfrac{p_i^{l+1/2} - p_i^{l-1/2}}{\Delta\widetilde{t}} + \dfrac{1}{K}\dfrac{p_i^{l+1/2} - \hat{p}_i^{l+1/2}}{\Delta t} = -\dfrac{[v_x]_{i+1/2}^l - [v_x]_{i-1/2}^l}{\Delta x} \\[4mm] \dfrac{1}{2\widetilde{G}}\dfrac{[\tau_{xx}]_i^{l+1/2} - [\tau_{xx}]_i^{l-1/2}}{\Delta\widetilde{t}} + \dfrac{1}{2G}\dfrac{[\tau_{xx}]_i^{l+1/2} - [\hat{\tau}_{xx}]_i^{l+1/2}}{\Delta t} = ... \\[4mm] \qquad\qquad = \left(\dfrac{[v_x]_{i+1/2}^l - [v_x]_{i-1/2}^l}{\Delta x} - \dfrac{1}{3}\dfrac{[v_x]_{i+1/2}^l - [v_x]_{i-1/2}^l}{\Delta x}\right) \\[4mm] \widetilde{\rho}\dfrac{[v_x]_{i+1/2}^{l+1} - [v_x]_{i+1/2}^l}{\Delta\widetilde{t}} = -\dfrac{(-(p_{i+1}^{l+1/2} - p_i^{l+1/2}) + [\tau_{xx}]_{i+1}^{l+1/2} - [\tau_{xx}]_i^{l+1/2})}{\Delta x}, \end{cases}$$ (B2)



## Appendix C: Matlab code

**Listing 1.** MATLAB Code for time loop computations

```matlab
1:  clear, figure(1), clf
2:  %physics
3:  Lx     = 1;
4:  w      = pi;
5:  u0     = 1;
6:  G0     = 1;
7:  K0     = 1*G0;
8:  %numerics
9:  nx     = 401;
10: nt     = 1e6;
11: K_G    = 1;
12: G      = 1;
13: K      = 1*G;
14: CFL    = 1/1.001;
15: %preprocessing
16: dx     = Lx/(nx-1);
17: x      = 0:dx:Lx;
18: for ipar = 1:25
19:     par     = 1+ipar/3;
20:     St      = par;
21:     dt      = 1;
22:     Vpdt    = dx*CFL;
23:     H       =  (K0+4/3*G0).*dt;
24:     dt_rho  = Vpdt.*Lx./St./H;
25:     Hdt     = Vpdt^2./dt_rho/1;
26:     Hr      = Hdt./(H);
27:     % initial
28:     V         = u0*sin(w*x/Lx);
29:     sigma       = 1*diff(V);
30:
31:     for it = 1:nt
32:         sigma       = (sigma + Hdt.*(diff(V)/dx   ))./(1 + Hr);
33:         V(2:end-1) = V(2:end-1)       + dt_rho*diff(sigma)/dx;
34:         if it == 1, mfun        = max(abs(V)) + max(abs(sigma));   end
35:         if (max(abs(V)) + max(abs(sigma))) < 1e-9*mfun,break;end
36:     end
37:     iters1(ipar) = it;    ipars1(ipar) = par;
38:     Maximum = iters1/nx; [find, find2] = min(Maximum(:)) ;
39: end
40: Reopt_num = ipars1(find2);
41: omega = 1; iparmax = 25;
42: for ipar = 1:iparmax
43:     par   = 1+ipar/3 ;
44:     St   = par;
45:     rho     = St^2*H^2/((1*K+4/3*G)*Lx^2);
46:     dt    = dx/sqrt((1*K+4/3*G)/rho);
47:     Vs    = sqrt((1*K+4/3*G)/rho); fun    = 1;
48:     for it = 1:nt
49:         fun_old=fun;
50:         a1 = pi^2*St*omega^2;
51:         a2= pi^2 + 1*St^2 ;
52:         a3= 2*St ;
53:         a4= 1;
54:         A = [a1 a2 a3 a4];
55:         Pol2        = roots( flip(A));         lambda       = max(real(Pol2));
56:         fun        = exp(  lambda(1)*Vs*it*dt ./Lx    ) ;
57:         if it == 1, mfun        = max(abs(  exp( lambda(1)*Vs*it*dt./Lx   ) ));   end
58:         err         = fun - fun_old;         merr        = max(abs(fun));
59:         if merr*6 < 1.0 *1e-9*mfun,break,end
60:     end
61:     iters(ipar)     = it;
62: end
63: figure(1);clf
64: plot(ipars1,iters1/nx,'-x','LineWidth',1 )
65: hold on; plot( 1+(1:iparmax)/3 ,iters/nx, 'o','MarkerSize',10,'LineWidth',1.5);  hold on;
66: Re_opt = 2*pi;
67: plot( [Re_opt Re_opt],[5 15],'LineWidth',3);hold on
68: plot( [Reopt_num Reopt_num],[5 15],'LineWidth',3); hold on
69: xlabel( '$ {\mathrm{St}}$ (-)','Interpreter','latex'  );
70: ylabel( '$ {n}_\textrm{iter}/n_x$ (-)','Interpreter','latex'  );
71: legend('Numerical solution','Analytical solution',...
72:     '$  {\mathrm{St}}_\textrm{opt} = 2  \pi $ ',...
73:     '${\mathrm{St}} - \textrm{numerical}$', 'Interpreter','latex');
74: grid on; drawnow
```





## Appendix D: Discretization: viscoelasticity

The discrete form of the system (27) can be written as

$$
\begin{cases}
\dfrac{1}{\widetilde{K}}\dfrac{p_i^{l+1/2}-p_i^{l-1/2}}{\Delta \widetilde{t}} + \dfrac{1}{K}\dfrac{p_i^{l+1/2}}{\Delta t} = -\dfrac{[v_x]_{i+1/2}^l-[v_x]_{i-1/2}^l}{\Delta x} \\[4mm]
\dfrac{1}{2\widetilde{G}}\dfrac{[\tau_{xx}]_i^{l+1/2}-[\tau_{xx}]_i^{l-1/2}}{\Delta \widetilde{t}} + \dfrac{1}{2G}\dfrac{[\tau_{xx}]_i^{l+1/2}-[\hat{\tau}_{xx}]_i^{l+1/2}}{\Delta t} + \dfrac{[\tau_{xx}]_i^{l+1/2}}{2\mu_s} = ... \\[4mm]
\qquad\qquad = \left( \dfrac{[v_x]_{i+1/2}^l-[v_x]_{i-1/2}^l}{\Delta x} - \dfrac{1}{3}\dfrac{[v_x]_{i+1/2}^l-[v_x]_{i-1/2}^l}{\Delta x} \right) \\[4mm]
\widetilde{\rho}\dfrac{[v_x]_{i+1/2}^{l+1}-[v_x]_{i+1/2}^l}{\Delta \widetilde{t}} = -\dfrac{(-(p_{i+1}^{l+1/2}-p_i^{l+1/2})+[\tau_{xx}]_{i+1}^{l+1/2}-[\tau_{xx}]_i^{l+1/2})}{\Delta x}.
\end{cases}
\tag{D1}
$$

## Appendix E: Discretization: quasi-static Biot's poroelastic equations

The discrete form of the system (44)-(46) can be written as

$$
\begin{cases}
\dfrac{1}{\widetilde{K}_1}\dfrac{[\bar{p}]_i^{l+1/2}-[\bar{p}]_i^{l-1/2}}{\Delta \widetilde{t}} + \dfrac{1}{K_u}\dfrac{[\bar{p}]_i^{l+1/2}-[\hat{\bar{p}}]_i^{l+1/2}}{\Delta t} = -\dfrac{[v_x]_{i+1/2}^l-[v_x]_{i-1/2}^l}{\Delta x} - B\dfrac{[q_x^D]_{i+1/2}^l-[q_x^D]_{i-1/2}^l}{\Delta x} \\[4mm]
\dfrac{1}{\widetilde{K}_2}\dfrac{[p_f]_i^{l+1/2}-[p_f]_i^{l-1/2}}{\Delta \widetilde{t}} + \dfrac{1}{K_u}\dfrac{[p_f]_i^{l+1/2}-[\hat{p}_f]_i^{l+1/2}}{\Delta t} = -B\dfrac{[v_x]_{i+1/2}^l-[v_x]_{i-1/2}^l}{\Delta x} - \dfrac{B}{\alpha}\dfrac{[q_x^D]_{i+1/2}^l-[q_x^D]_{i-1/2}^l}{\Delta x}
\end{cases}
,
\tag{E1}
$$

$$
\begin{cases}
\dfrac{1}{2\widetilde{G}}\dfrac{[\overline{\tau}_{xx}]_i^{l+1/2}-[\overline{\tau}_{xx}]_i^{l-1/2}}{\Delta \widetilde{t}} + \dfrac{1}{2G_u}\dfrac{[\overline{\tau}_{xx}]_i^{l+1/2}}{\Delta t} = \left( \dfrac{[v_x]_{i+1/2}^l-[v_x]_{i-1/2}^l}{\Delta x} - \dfrac{1}{3}\dfrac{[v_x]_{i+1/2}^l-[v_x]_{i-1/2}^l}{\Delta x} \right)
\end{cases}
,
\tag{E2}
$$

$$
\begin{cases}
\widetilde{\rho}_t\dfrac{[v_x]_{i+1/2}^{l+1}-[v_x]_{i+1/2}^l}{\Delta \widetilde{t}} = -\dfrac{(-([\bar{p}]_{i+1}^{l+1/2}-[\bar{p}]_i^{l+1/2})+[\overline{\tau}_{xx}]_{i+1}^{l+1/2}-[\overline{\tau}_{xx}]_i^{l+1/2})}{\Delta x} \\[4mm]
\widetilde{\rho}_a\dfrac{[q_x^D]_{i+1/2}^{l+1}-[q_x^D]_{i+1/2}^l}{\Delta \widetilde{t}} = -[q_x^D]_{i+1/2}^l - \dfrac{\frac{k}{\eta_f}([p_f]_{i+1}^{l+1/2}-[p_f]_i^{l+1/2})}{\Delta x}
\end{cases}
.
\tag{E3}
$$

*Author contributions.* YA designed the original study, developed codes and algorithms, performed benchmarks, created figures and edited the manuscript. YYP provided early work on accelerated PT methods, contributed to the original study design, developed codes and algorithms, helped out with the dispersion analysis, edited the manuscript, supervised the work.

*Competing interests.* The contact author has declared that none of the authors has any competing interests.

*Acknowledgements.* We thank Ivan Utkin and Lyudmila Khakimova for stimulating discussions.



*Financial support.* Yury Alkhimenkov gratefully acknowledges the support of the Swiss National Science Foundation (project number P500PN_206722).

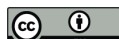



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
