# Peer review of "Accelerated pseudo-transient method for elastic, viscoelastic, and coupled hydro-mechanical problems with applications"

_Geoscientific Model Development, 2024_

## Author Comment (AC1)

This manuscript delves into the utilization of the Accelerated Pseudo-Transient (APT) method for tackling quasi-static elastic, viscoelastic, and coupled hydro-mechanical problems. The study not only derives but also rigorously tests the numerical APT formulations tailored for these specific problem sets. Introducing novel dimensionless parameters (St and I1, I2) for the APT method in the context of elastic and coupled poroelastic equations marks a notable advancement. The manuscript showcases the efficacy and adaptability of the proposed APT method through high-resolution 2D and 3D nonlinear modeling results. These simulations vividly illustrate the method's flexibility and efficiency in handling complex geoscience scenarios. This contribution of the APT method to the modeling of realistic geoscience problems is significant and warrants publication in GMD.

While recognizing the manuscript's importance, I acknowledge that certain sections suffer from unclear or confusing descriptions, likely stemming from the writing style and flow. Therefore, I recommend substantial revisions to enhance clarity and coherence throughout the manuscript. This includes addressing the major modifications outlined and attending to various smaller edits that may be necessary for improved readability and comprehension.

Thank you for your thorough and constructive feedback on our manuscript. We are very grateful for your recognition of the significance of our work, especially the introduction of novel dimensionless parameters (St and I1, I2) in the context of elastic and coupled poroelastic equations, as well as the validation of the Accelerated Pseudo-Transient (APT) method for complex geoscience problems.

We fully acknowledge your concerns regarding the clarity and flow of certain sections, and we take this feedback seriously. We are committed to revising the manuscript to enhance both clarity and coherence. We will carefully address the major modifications you've highlighted and ensure that the smaller edits necessary for improved readability are also attended to.

Our revisions will focus on refining the descriptions and improving the overall writing style to ensure that the important concepts, methodologies, and results are communicated more effectively. Once completed, we are confident that the manuscript will better reflect the rigor and importance of the work, as well as meet the high standards of *GMD*.

Thank you again for your valuable insights.

Sincerely,
Yury Alkhimenkov and Yury Podladchikov

Below are the comments and edits from my sides, with bold text for the major ones.

**Line 90-95. and 100-105   This description of 1ˢᵗ order and accelerated PT method is not clear or correct.     converges to 0, suggest vx to 0.  It does not make sense.   For APT,  you should involve 2ⁿᵈ derivative of Vx like  in Eq.6) and Eq.7) of Rass 2022, since you cite it.  But it is  not clearly stated.   Correct this!**

We appreciate the reviewer's comments regarding the different versions of the pseudo-transient method. This section is rewritten and corrected. Correct references were added.

**Line 225:    "Naïve" does not sound good here!    " that there are minimal modifications to the original formulation of" is not a good description for this scheme.   I think "Elegant APT scheme " has even smaller modifications (only refine G).   Clarify this!**

We agree with the reviewer's suggestion and have revised the text accordingly. We have retained the phrase "APT scheme" in the main text while removing the word "elegant."

**In fact, I think "Naïve APT scheme"  part can be removed. It is just complicated but not naïve! It added only confusion to your description.  It is a natural transition from the scheme of elastic equation to the viscoelastic equation (Eq. 26).**

We have removed the title "Naïve scheme" and added further explanation regarding its potential applications to Appendix. Additionally, we have removed the term "Naïve" to maintain a more formal scientific tone.

**Line 340: As I wrote above, the formulation in 4.14 is needed in 4.1.3.  Perhaps you can do some adjustment.**

We understand the reviewer's concerns regarding Section 4.1.3. The reason we initially provided the formulation only in Section 4.1.4 is that the decoupled equations yield simpler roots that fit more neatly on the page. In contrast, the full equations for the roots are much more complex, with numerous additional cross terms, which is why we included Maple routines to validate this part.

In response to the reviewer's comments, we have moved Section 4.1.4 earlier, making it the new Section 4.1.3, and added more explanation to clarify the equations. Additionally, we have improved the Maple files to better address the reviewer's concerns. As a result, Section 4.1 has been significantly improved in the current version of the manuscript.

**Line 482.  I am not convinced about the sensitivity of optimized numerical parameters on boundary conditions from your example. You need more tests to convince people.**

 We agree with the reviewer that additional examples are necessary to demonstrate that optimized numerical parameters depend on boundary conditions. After extensive investigation, we have updated our statement to: "There is only moderate sensitivity of optimal numerical parameters with respect to boundary conditions." We found that the sensitivity to initial conditions and non-linearities is much more pronounced. We have improved the explanation in this section to reflect these findings.

Line 6;  replace "manuscript"  with "study".

Done!

study

Line80-85    Eq. 4)     I recommend to write it to $\sigma_{xx}$- $\sigma_{xx}$ _old//dt  to clarify.  The it is similar to the 1$^{st}$ ord PT method case in rass 2022.   Clarify that you aim to solve one transient step for this time-dependent problem. Otherwise, it is quite confusing!

We have modified this section and emphasized in the manuscript that the PT method can be used to solve elasticity equations for calculating effective elastic properties. In this case, $\sigma xx$ _old  is zero, allowing us to solve the simplified equations. However, in response to the reviewer's request, we have further improved the explanation in this section to provide greater clarity. We removed the reference to Rass et al since in our study we have elasticity (elliptic) equation and Rass et al did not present this.

Line 90-95  100-105.  Eq.5) and 6) the same as Eq.4)!   Do it like in Eq.7): do  time (real physic) discretization of $\sigma_{xx}$.

We have t with tilde (which correspond to the pseudo-time) and t without tilde (which correspond to the physical time). The way we present equations are the standard in geophysical (mechanical, math) literature for time partial derivatives.

Line 104.  To avoid confusion: " Propagating waves in pseudo physical space."

We improved the wording in the manuscript.

 (i) Inertial terms are added into the constitutive relations, (ii) Inertial terms are responsible for wave propagation in pseudo physical time and space (i.e., hyperbolic) and viscous terms (treated as a Maxwell rheology) are the physical quantities.

Line 109:   "into the equation stress"  is not clear!   Remove "stress"?

We improved the explanation.

Inertial terms are added into the constitutive relations

Better description is need for "(ii) these terms are treated as a Maxwell rheology (a viscous damper)". As I understand, Eq.7a) use a maxwell model of rheology , the item σ_xx/Δt as a viscous part; while the pseudo item is the elastic part.

The reviewer is completely correct and we improved the explanation.

Here we report a modification of the APT method. The solution of the quasi-static elasticity equations can be achieved in two steps. (i) Inertial terms are added into the equations constitutive relations, (ii) Inertial terms are responsible for wave propagation in pseudo physical time and space (i.e., hyperbolic) and viscous terms (treated as a Maxwell rheology) ate the physical quantities.

Line 112:     What is the reason to choose =H? Is there a better choice?  You said   is to be determined.  Perhaps would also has an optimal choice.

The reason for our approach is simplicity. This equation involves only one numerical parameter, $St$, while the other parameters are dependent. If we used a different value for $\tilde{H}$ instead of $H$, we would need to modify the entire numerical scheme and adjust the $St$ value, without any improvement in convergence, as we are constrained by the CFL condition and the single numerical parameter $St$. There is only one degree pf freedom = one parameter.

Line 115:     It is not good to say Eq.7) can be simplified to Eq.8), which could change the equation.  But  I know σ_xx_old as a constant can be ignore for  the derivation process.  Please write better description for it.

The reviewer is completely correct and we improved the explanation.

For the analysis of the system of equations \eqref{dve_14} we can omit $\hat{ \sigma}$ since the stress $\hat{ \sigma}$ does not change inside the loop over ``pseudo" time $\widetilde{t}$:

Line 142:   How about "Instead, the following combinations are needed  for the numerical implementation of the APT algorithm."?

The reviewer is completely correct and we improved the explanation.

Instead, the following combinations are needed for the numerical implementation of the APT algorithm

Line 145-146. Notice "f" is already use as the function name before you  write "f is the frequency"

The reviewer is completely correct and we modified the variable name.

```
\begin{equation}\label{eq_111111}
F(\widetilde{t}, x) = \exp{ \left[  \dfrac{( \gamma \, \widetilde{V}_p\,  \widetilde{t} + \pi
\,\omega \,x \,i)}{L_x}  \right]},
\end{equation}
```

Line 157.   Is  "minimum"  suitable here ? Line 158  "This minimum reaches maximal value" is confusing…

The reviewer is completely correct and we improved the explanation.

The real parts of the roots $\gamma_1$ and $\gamma_2$ control the exponential decay rate of the solution \citep{rass2022assessing}, therefore, we are interested in the minimum of these values. This minimum reaches its value when the discriminant is zero:

Line 186.   Fig. 1 show that damping scheme 2 generate different stress with scheme 1.    Why? You did not talk about it in section 2.3.4

 In response to the reviewer's request, we updated the explanation of this section and removed scheme 1 from the main text. The reason for different stress was that scheme 1 (in the previous notation) was not fully correct.

Fig3.  There are two subplots, but there is no description of it, neither in the caption or in the main text.

 The reviewer is completely correct and we added the explanation and reference in the main text.

It can be seen that the analytical and numerical results are in excellent agreement (Figure \ref{FigVS1}) that validates the proposed approach.

Line 290.   It would be nice to clairfy the (pseudo ) physical meaning of I2.

We improved the explanation of I2. It actually has physical meaning in the framework of poroelasticity.

The physical meaning of $I_2$ is the following: $I_2$ controls the behavior of the Biot's slow wave, if $I_2 \ggg 1$ the slow wave behaves as a propagating wave, if $I_2 \lll 1$ the slow wave behaves as a diffusive mode.

Line 300.  Need a bit explanation on the choice of numerical parameter K1=K_u  G1=Gu.

The reason for our approach is simplicity as in the previous section. We added some explanation into the manuscript.

The reason for setting $\widetilde{K}_1={K}_u$ and $\widetilde{G}_1={G}$ is simplicity, since the 4-th order equation has only two degrees of freedom, a different choice of these parameters would simply re-scale the two final optimal parameters.

Line 299 and 335.   How come the optimized St value is  St=2*pi and St=2.9?  formulation?   From Fig.5, I can see you do have a formulation.  It would be nice to write it down in the main text or appendix.

 The values came from analytical derivations using the Maple file. The resulting values are the roots of the equations. We improved the explanation in the corresponding section. The full equations are very cumbersome and we advise to consult the Maple file for an interested reader.

Fig. 6. The caption is too cumbersome with a lot of repetition. Simplify it!

The reviewer is completely correct and we shortened and improved the explanation.

\caption{Convergence rate in a homogeneous poroelastic medium for different $I_2$: numerical result as a function of the dimensionless parameter ${\mathrm{St}}$. Panel (a): $I_2 =  100$. Panel (b): $I_2 = 0.01$. Panel (c):  $I_2 = 100$. Panel (d) $I_2 = 0.01$.
}

Fig. 6. For the 3D case, the optimized St are 28 for both I2=100 and I2=0.01, while they are different for 1D and 2D. Explain it!

We revised this section and deleted 3D results. We analyze 3D results in the discussion section.

 \subsubsection{2D numerical simulations: poroelasticity}

The accuracy of the proposed ${\mathrm{St}}_\text{opt}$ is illustrated numerically in 2D (Figure~\ref{Fig_2Dv1}a-b). It can be seen that the results presented here for 1D need some calibration to be applied to 2D simulations. Note that the numerical parameters are sensitive to boundary and initial conditions, which is explored below. Therefore, some test must be performed for each numerical setup.

Line 400.   Without comparison of low resolution, I can not see the thickness of shear band is mesh-independent.

 The reviewer is correct and to prove it, we would need even higher resolution for comparison. However, without regularization the localization of shear bands is 1 grid cell. In our implementation we have several grid cells (more than 10) which already proves the mesh-independency of our simulation according to the study by

Resolving strain localization in frictional and time-dependent plasticity: Two-and three-dimensional numerical modeling study using graphical processing units Y Alkhimenkov, L Khakimova, I Utkin, Y Podladchikov

We added the references.

Figure~\ref{Press_2D_HR1} shows the results of the 2D simulation with an ultra-high resolution of $N=10,239^2$ grid cells. The finite thickness of the shear bands confirms that the simulation is mesh-independent as it has been shown by \cite{https://doi.org/10.1029/2023JB028566}.

Line 450.   Here you say St_opt=2*pi*sqrt(3).    It is different with Fig. 6 (28).   A lot is missing.  Perhaps you should provide 2D and 3D derivation process. I could not find it in the maple file.

We did not make available the rigorous derivation process in 2D and 3D in Maple for a public. We agree that the values are different from Fig 6 and we modified this section. Indeed, as a first guess St_opt=2*pi*sqrt(3) but his value should be adjusted with respect to boundary and initial conditions and nonlinearities involved. See new discussion section.

Fig.11.   Please put boundary conditions information on the subtitle of b and c.  it would made the figure more readable!

Done.

Line 469.  "highly sensitive"?      The change is only from 4.63 to 6.0142.  It is not very sensitive. You need another example to say it is highly sensitive!

The reviewer is completely correct, and as we mentioned earlier, we have revised this conclusion. The sensitivity is only minor.

Lawrence H.Wang

We would like to thank the reviewer again for valuable comments, which helped us improve the quality of the manuscript.

Sincerely,
Yury Alkhimenkov and Yury Podladchikov

---

## Author Comment (AC2)

Response to Reviewer 2: Our comments are provided in blue. Text modifications are provided in green.

The the Accelerated Pseudo-Transient (APT) method is a matrix-free approach for iteratively solving partial differential equations (PDEs) which is embarrassingly parallel, thus being highly suitable for GPUs. The main challenge of the APT is to fine-tune the numerical parameters it introduces in the PDEs to obtain the optimal convergence rates.

In this paper the authors present a comprehensive analysis of the APT equations for quasi-static elastic and viscoelastic equations, and coupled hydro-mechanical problems, showcasing the derivation of the corresponding optimal numerical parameters. The manuscript highlights the accuracy and robustness of the APT to handle 2/3D highly-non linear coupled problems, as well as demonstrating the capability of the APT to reach extremely high resolutions.

I believe the outcome of the manuscript is relevant and is worth of a GMD publication. However, the manuscript requires of some major improvements before publication to largely improve its clarity and readability. Below is a detailed list of major and minor comments.

**General comments**

- I feel like the manuscript is lacking of many details that are either missing or should be explained in more detail and in a clear way; line by line comments below. Some sections manuscript (e.g. introduction) would also largely benefit of some rewriting to improve the clarity and quality of the text.

We would like to thank the reviewer for highlighting the need for improvements in clarity and readability. We agree that certain sections of the manuscript would benefit from further explanation. As such, we have rewritten parts of the article to ensure the content is more accessible and comprehensible to the readers.

- Perhaps I am missing something, but I don't think it is obvious what is the numerical problem being solved in

 - Section 2.3.4 / Figure 1

 - Section 2.3.6 / Figure 2

 - Figure 3

 - Section 4.1 / Figure 4

 - Section 4.1.5 / Figure 6

 Some clarification may help. Furthermore, Figure 3 seems not to be referenced /

discussed in the manuscript; and it also has two sub panels that are not described in the the caption neither.

We agree with the reviewer that some more explanation is needed. In all the figures, a comparison between analytical and numerical solutions is presented. We added some explanation before figure 1.

\paragraph{Problem statement: validation of the numerical parameters}\label{pr}

To validate the numerical parameters, the following experiment is performed: in the numerical solver, we set all boundary conditions to zero and initialize the system with a sinusoidal wave. The numerical solution is then run over pseudo-time until it converges to a specified precision (i.e., $10^{-12}$). Simultaneously, the same equation is solved using the analytical method (amplification matrix) to achieve the same precision (i.e., $10^{-12}$). The results are then compared as a function of $\mathrm{St}$. Ideally, the results should be identical or very close, which would validate the choice of numerical parameters and the applied numerical scheme. For the numerical solution, we use a classical conservative staggered space-time grid discretization \citep{virieux1986p} which is equivalent to a finite volume approach \citep{dormy1995numerical}. More details on the present discretization can be found in \cite{alkhimenkov2021resolving, alkhimenkov2021stability}.

- I encourage the authors to use the colormaps available either in the _PerceptualColourMaps_ package or in Fabio Crameri's _Scientific Colour Maps_. Both set of colormaps are available in MATLAB.

We agree with the reviewer there are other colormaps exists. We use standard colormap in Matlab "jet" as we have used in all our previous articles. There is no strict requirement on the colormap choice, therefore, we keep jet colormap. We may consider in the future to use other colormaps.

- I would not consider MATLAB being truly open-sourced as a license needs to be purchased. It is true that most of the (at least European) universities have institutional licenses, but not all the readers interested in trying out the scripts provided here may have access to a license. For this reason I would also like to encourage the authors to consider using other free dynamic languages, such as Julia or Python, for future work/publications.

We agree with the reviewer that Matlab is not open-access. However, there is an alternative --- Octave which is open access. The results presented in .m files can be reproduced using Octave.

- Attached is a pdf with other comments and other typos/grammatical corrections.

**Line by line**

*L15/62* Voxels do not exist in 2D, they are called pixels, which are 2D bitmaps. Either way, the domain of a 2/3D simulation is discretised in cells or elements. Please replace "voxels" with "cells", "elements" or similar throughout the manuscript.

We agree with the reviewer. This depends on the community. In computational mechanics, scientist call grid cells and elements as voxels. We replaced voxels into grid cells as suggested.

grid cells

*L25/26* The APT actually relies quite a bit on storage of data on matrices, as the iterative solver needs to be split into several kernels to avoid race conditions. The actual advantage of matrix-free methods is that they avoid assembling a global sparse matrix and either expensive direct solves or other iterative methods that rely on not-so-cheap sparse matrix-vector multiplications.

We agree with the reviewer. The APT method **is local** and matrix-free in a sense that we do not need a global matrix as in direct solvers. APT is free from global scalar products that involve information from full arrays (as in conjugate method).

The Accelerated Pseudo-Transient (APT) method is designed to iteratively solve a modified version of the original partial differential equation (PDE) by introducing inertial and relaxation terms. This modified PDE is repeatedly solved until the added pseudo-physical terms vanish, providing an accurate approximation of the solution to the original equation. The APT method becomes increasingly efficient when implemented with exclusively spatially local operations, eliminating the need to access global storage for evolving fields. Unlike the conjugate gradient method, which requires two global scalar products per iteration, the APT method advances without global memory operations, enhancing computational performance by utilizing fast cache memory.

*L30* effectively => efficiently

Corrected!

efficiently

*L35* This whole paragraph would largely benefit of some rewriting, it reads as a collection of facts without any flow. I would also say that the first sentence can be easily removed as it does not bring anything to the topic of APT.

We agree with the reviewer that this paragraph may benefit from some rewriting. We think that this paragraph provides a general overview of the development of PT methods in chronological order. Also, the first sentence is importance since it reference one of the first iterative methods to solve PDEs which we describe on the paper.

One of the first pseudo-transient (PT) iterative methods to solve elliptic PDEs was presented by \cite{richardson1911ix}. An improved PT method for elliptic problems, which can be referred to as the Accelerated Pseudo-Transient (APT) method, was proposed in the 1950s by \cite{frankel1950convergence} and further investigated by \cite{riley1954iteration} and \cite{young1972second}. The pseudo-transient method is also known as a dynamic-relaxation (DR) method that was used by \cite{otter1965computations, otter1966dynamic}…

*L70* I don't think $nabla dot$ is an operator itself, it just means the dot product of the nabla operator and something else. The authors should also remove the references regarding the nabla operator, as this notation has been introduced and widely much earlier (by Hamilton in the 1800s) than in those references and it is a widely known, accepted, and used notation. If you want to keep the mathematical definition of nabla, define it when you introduce the symbol.

We agree with the reviewer that there are different interpretations. Some scientists refer to the $\nabla \cdot$ (divergence) as an operator. Regardless, the statement is clear and not open to misinterpretation.

*Eq2* Since tensor notation is being used, I suggested the authors to denote the rates using the dot notation instead, i.e. $dot(epsilon)$

We agree with the reviewer that there are other ways to denote rates. The derivative of a tensor (via \dot) may reflect partial derivative, full derivative of material derivative or objective (e.g., Jaumann derivative). To make our statement clear we keep partial derivative to separate from other possible choices.

*Eq3* The tensor products should be dropped, it is $dot(epsilon) = 1/2(nabla bold(v) + (nabla bold(v))^T)$

We agree with the reviewer that there are other ways to write this equation. Eq3 is correct with and without tensor products. The gradient of a vector field is the same as the dyadic product of the del operator and the vector.

We refer to the standard terminology in micromechanics, see

"Micromechanics: overall properties of heterogeneous materials. S Nemat-Nasser, M Hori" or "Introduction to micromechanics and nanomechanics. S Li, G Wang"

*L79* superscript T

Corrected.

*Section 2.3* Perhaps it is a good idea to expand a bit on the pseudo transient method, rather than directly writing down the equations. It may not be obvious for the general reader to know what's going on. You could for example explain that the equations are written in their residual form and the pseudo time derivatives are added to the left hand side (or wherever you write down the zero), which should vanish upon convergence, thus recovering the original equations; or similar.

We agree with the reviewer that some more explanation might help. The original text contains the sentence: *The main idea is that the solution of a quasi-static equation (stationary process), usually described by an elliptic PDE, is represented by an attractor of a transient process described by parabolic or hyperbolic PDEs.* We added more explanation into the introduction and corresponding section:

The Accelerated Pseudo-Transient (APT) method is designed to iteratively solve a modified version of the original partial differential equation (PDE) by introducing inertial and relaxation terms. This modified PDE is repeatedly solved until the added pseudo-physical terms vanish, providing an accurate approximation of the solution to the original equation. The APT method becomes increasingly efficient when implemented with exclusively spatially local operations, eliminating the need to access global storage for evolving fields. Unlike the conjugate gradient method, which requires two global scalar products per iteration, the APT method advances without global memory operations, enhancing computational performance by utilizing fast cache memory. This method is versatile, applicable to both linear and nonlinear equations, and distinguishes itself with several key attributes. (i) APT is a matrix-free method, enabling the solution of large-scale 3D problems without the overhead of matrix storage. (ii) leveraging only local operations, APT naturally lends itself to parallelization, making it well-suited for modern computing architectures. (iii) its structure facilitates efficient implementation on Graphical Processing Units (GPUs), capitalizing on their ability to handle parallel tasks efficiently. (iv), APT method aligns closely with the physics of wave phenomena, offering a robust theoretical framework for rigorous understanding and application.

Simply put, the equations are written in their residual form, and pseudo-time derivatives are added to the left-hand side. The solution is achieved once the pseudo-time derivatives attenuate to a certain precision (e.g., $10^{-12}$).

*L87* system of equations; in plural, this mistake is repeated several times, please correct it everywhere.

Corrected!

system of equations

*L102* Please define $tilde(rho)$ as well

Corrected.

where $\mu$ and $\widetilde{\rho}$ are the damping parameters.

*L104* compare =>compared

Corrected

compared

*L109* equation stress => constitutive equation

Corrected

constitutive equation

*L112* Is $tilde(H)$ really equal to $H$? How did you reach to this conclusion?

The reason for our approach is simplicity. This equation involves only one numerical parameter, \(St\), while the other parameters are dependent. If we were to use a different value for \(\tilde{H}\) instead of \(H\), we would need to modify the entire numerical scheme and adjust the \(St\) value, without any improvement in convergence, as we are constrained by the CFL condition and the single numerical parameter \(St\).

*L115/120* When the reader reaches line 115, it is not obvious why the stress from the previous time step suddenly vanishes. The authors should explain here why this happens, rather than doing it later on.

We added a general description saying that there are two for loops – one is physical time (related to loading) and inner loop is in "pseudo-time".

For the analysis of the system of equations \eqref{dve_14} we can omit $\hat{ \sigma}$ since the stress $\hat{ \sigma}$ does not change inside the loop over ``pseudo" time $\widetilde{t}$:

…

\subsubsection{Problem statement}

The system of equations \eqref{eq:1}-\eqref{eq:2} can be applied to solve many problems in solid mechanics. Particularly, as an example in this study, we use these equations to solve two applied problems: (i) - loading/unloading of an elastic body and (ii) - calculation of effective elastic properties.

For the analysis of loading/unloading processes in an elastic body, the system of equations \eqref{dve_1} is discretized with a physical time step $\Delta t$, which is intrinsically linked to specific strain increments.
The loading/unloading process is simulated through a series of time increments, cumulatively spanning the total time of interest.
This total time corresponds to the overall strain accumulation within the elastic body. In contrast, when computing effective elastic properties (task ii), the system of equations \eqref{dve_1} is utilized with a single loading increment, characterized by a physical time step $\Delta t$.
This solitary increment corresponds to a single strain loading step.
Subsequently, the stress and strain fields are spatially averaged across the model domain.
The division of these averaged quantities yields the effective elastic moduli.

*L122* provided in Appendix A. A discrete => is provided in Appendix A, and a discrete...

Corrected

The APT version of expression \eqref{dve_14} (or \eqref{dve_141}) where the stress tenor is decomposed into pressure and deviatoric stress tensor is provided in Appendix \ref{Ap00}, and a discrete version of the system \eqref{dve_141} is provided in Appendix \ref{Ap1}.

*L136* calculated => defined

Corrected

defined

*eq11* why not using normal brackets for the exponential instead of straight brackets? should be clear enough

We agree with the reviewer that there are several options possible. This is a notation choice. We keep the present notation.

*L146* $exp$ is standard notation and needs no definition, please remove from the manuscript. It is also written later on in the manuscript.

We agree with the reviewer that there are several options possible. This is a notation choice. We keep the present notation. We removed this from the manuscript (second time) which is written two times.

*L147* I am not familiar with the concept of amplification matrix. Could the authors briefly comment on it?

We have added a reference book dealing with stability of discrete numerical schemes and using this terminology.

This is a standard procedure used for example in determining the correct CFL condition. It is well explained in many text books, for example, in Hirsch (1988).

See also Stability of discrete schemes of Biot's poroelastic equations

Y. Alkhimenkov ,1,2,3 L. Khakimova 3,4 and Y.Y. Podladchikov

\citep{hirsch1988numerical, alkhimenkov2021stability}

*Section 2.3.4* I am afraid I am bit lost here. Could the authors please elaborate and provide some more details of what is actually being solved here, and what exactly are the numerical and analytical solutions?

We added some explanation into this section.

\paragraph{Problem statement: validation of the numerical parameters}\label{pr}

To validate the numerical parameters, the following experiment is performed: in the numerical solver, we set all boundary conditions to zero and initialize the system with a sinusoidal wave. The numerical solution is then run over pseudo-time until it converges to a specified precision (i.e., $10^{-12}$). Simultaneously, the same equation is solved using the analytical method (amplification matrix) to achieve the same precision (i.e., $10^{-12}$). The results are then compared as a function of $\mathrm{St}$. Ideally, the results should be identical or very close, which would validate the choice of numerical parameters and the applied nimerical scheme.

*Section 2.3.5* The authors should briefly explain (here or elsewhere in the main body of the manuscript) that the equations are discretised with a staggered grid and finite difference scheme. This is only mentioned in the appendix.

We added some explanation into this section.

For the numerical solution, we use a classical conservative staggered space-time grid discretization \citep{virieux1986p} which is equivalent to a finite volume approach \citep{dormy1995numerical}. More details on the present discretization can be found in \cite{alkhimenkov2021resolving, alkhimenkov2021stability}.

*Figure 1* I'm guessing (-) means that there are no units. This symbol could be removed from the axis labels if you state in the caption that everything is dimensionless. I also suggest the authors to put the name of the field (e.g. Vx) in the y-axis of the plots, instead of putting it in the title and writing Amplitude. These comments apply to all the plots.

We agree with the reviewer that there are several representations can be chosen. In our opinion the present representation is clear.

Why the stress is about 4 orders of magnitude different between scheme 1 and 2?

In response to the reviewer's request, we updated the explanation of this section and removed scheme 1 from the main text. The reason for different stress was that scheme 1 (in the previous notation) was not fully correct.

*L190* The boundary conditions could be expressed as function of the spatial coordinate ($v_x (x=0)=1$ and $v_x (x=L_x)=0$) instead of nodal numbering. In this way they have a physical meaning and would simplify this sentence in the manuscript.

We agree with the reviewer that there are several representations can be chosen. In our opinion the present representation is clear.

*L199* I think it is more clear if the accuracy is expressed as residuals instead of pseudo time derivatives

We agree with the reviewer. We express now in resudials.

After $5 \, n\_x$ iterations in ``pseudo-time" we can report the accuracy (in residuals) $d v\_x = 10^{-13}$. This result correspond to the difference between the numerical value for $H^*$ and the analytical value for $H^*\_{an}=7/3$ via $(H^*\_{an}-H^*\_{num})/H^*\_{an}\times 100\% $ to as $10^{-12}\%$.

*Section 2.3.6* As in Section 2.3.4, please add more details of what is being solved.

We added some explanation in the beginning of the paper.

\subsubsection{Problem statement}

The system of equations \eqref{eq:1}-\eqref{eq:2} can be applied to solve many problems in solid mechanics. Particularly, as an example in this study, we use these equations to solve two applied problems: (i) - loading/unloading of an elastic body and (ii) - calculation of effective elastic properties.

For the analysis of loading/unloading processes in an elastic body, the system of equations \eqref{dve\_1} is discretized with a physical time step $\Delta t$, which is intrinsically linked to specific strain increments.
The loading/unloading process is simulated through a series of time increments, cumulatively spanning the total time of interest.
This total time corresponds to the overall strain accumulation within the elastic body. In contrast, when computing effective elastic properties (task ii), the system of equations \eqref{dve\_1} is utilized with a single loading increment, characterized by a physical time step $\Delta t$.
This solitary increment corresponds to a single strain loading step.
Subsequently, the stress and strain fields are spatially averaged across the model domain.
The division of these averaged quantities yields the effective elastic moduli.

\paragraph{Problem statement: validation of the numerical parameters}\label{pr}

To validate the numerical parameters, the following experiment is performed: in the numerical solver, we set all boundary conditions to zero and initialize the system with a sinusoidal wave. The numerical solution is then run over pseudo-time until it converges to a specified precision (i.e., $10^{-12}$). Simultaneously, the same equation is solved using the analytical method (amplification matrix) to achieve the same precision (i.e., $10^{-12}$). The results are then compared as a function of $\mathrm{St}$. Ideally, the results should be identical or very close, which would validate the choice of numerical parameters and the applied numerical scheme. For the numerical solution, we use a classical conservative staggered space-time grid discretization \citep{virieux1986p} which is equivalent to a finite volume approach \citep{dormy1995numerical}. More details on the present discretization can be found in \cite{alkhimenkov2021resolving, alkhimenkov2021stability}.

*Section 2.3.7* I assume the boundary conditions and resolution are as in 2.3.5, but please clarify it in the text.

We added some explanation into this section.

Let us again consider a 1D numerical domain with $L_x=1$, which is discretized into $n_x=1000$ grid cells. The boundary conditions are the same as in the previous section 2.4.2. (Numerical experiment 2). Now, we consider a heterogeneous medium in 1D represented by layers of different elastic properties.

*L207* We perform *the* numerical

Corrected.

We perform the numerical experiment

*L211* I assume $phi$ is the volume fraction of the weakest phase? please clarify in the text

Corrected.

where $A$ is a minimum of the elastic moduli of the softest material divided by volume fraction of the weakest phase $\phi$:

*eq 25* Were other setups tested? Dos this still work $K$ and $G$ are very different?

In this study we did not explore all possible scenarios. In the text: *Note that the definition of A in equation (25) is valid for the specific parameters of the medium considered here and is not universal.*

*L203* Figure Figure 2 => Figure 2

Corrected.

*L215* The authors should explain how is this accuracy defined, as now it appears as a percentage while in the previous sections it was the value of the residual. It would also help to understand why the value for scheme 1 is much larger than for the scheme 2.

We added some explanation on the accuracy definition. We removed scheme 1 from the main text and added it into appendix (a corrected version).

After $5 \, n_x$ iterations in ``pseudo-time" we can report the accuracy (in residuals) $d v_x = 10^{-13}$. This result correspond to the difference between the numerical value for $H^*$ and the analytical value for $H^*_{an}=7/3$ via $(H^*_{an}-H^*_{num})/H^*_{an}\times 100\%$ to as $10^{-12}\%$.

*Section 3* In the previous sections the authors were using tensor notation to describe the system of equations. For consistency, it would be great if all the systems of equations presented here were using the same notation.

We modified the previous sections and added component notations as well. In the present section 3, we added the full set of viscoelastic equations.

Now, let us consider viscoelastic equations. The general form is the following:
\renewcommand*{\arraystretch}{2}
\begin{equation}\label{dve_12VE0}
\left\{
\begin{array}{ll}
   \dfrac{1}{K} \dfrac{ \partial p }{\partial t} = - \nabla \cdot \mathbf{v} \\
   \dfrac{1}{2G} \dfrac{\partial \boldsymbol{\tau}}{\partial t} + \dfrac{\boldsymbol{\tau}}{2 \mu_s} = \boldsymbol{\varepsilon} - \dfrac{1}{3} (\nabla \cdot \mathbf{v}) \mathbf{I}_2\\
    {0} = \nabla \cdot (-p \mathbf{I}_2 + \boldsymbol{\tau}) ,
\end{array}
\right.
\end{equation}
where $\mu_s$ is the shear viscosity of the solid material, $p$ is the pressure, $\boldsymbol{\tau}$ is the deviatoric stress tensor ($\sigma = - p \mathbf{I}_2 + \boldsymbol{\tau}$.

*L223* (physical) viscosity => shear viscosity

Corrected.

=> shear viscosity

*Figure 3* If I am not mistaken, this figure is not referenced or discussed in the manuscript.

We agree with the reviewer. Yes, indeed. We added references and explanations to Fig. 3.

It can be seen that the analytical and numerical results are in excellent agreement (Figure \ref{FigVS1}) that validates the proposed approach.

*Section 3.2* I do not find the name of the section appropriate, as "elegant" is a rather subjective and arbitrary term and there are only some minor changes w.r.t the previous subsection

We agree with the reviewer. Yes, indeed. This section is revised reflecting the present comments and the comments from the reviewer 1.

*eq 46* The left hand side can be simplified

$mat(

 tilde(rho)_t (partial v_i ^s) / (partial tilde(t));

-tilde(rho)_a (partial q_i ^D) / (partial tilde(t));

)$

The purpose of having the full matrices is to highlight that there are no added mass coefficients in off-diagonal components as in Biot's equation (see eq 8 in

Resolving wave propagation in anisotropic poroelastic media using graphical processing units (GPUs). Y Alkhimenkov, L Räss, L Khakimova, B Quintal, Y Podladchikov

*L319* These coefficients have already been defined. And please remove the definition of $exp$.

We removed repetitions and definition of exp.

*Sections 4.1.2 / 4.1.3* As before, explain what is being solved

We added some explanation into the text.

(see explanation in section~\ref{pr})

*Figure 6* If I didn't miss anything, the $"St"_("opt")$ for the 3D case is much larger than any of the values described in the text. Does this mean that the only way to tune this parameter in the 3D case is trial and error?

This is a good point and we spend some time on further defining the correct St for 3D. Still, analytical estimation of St gives a reasonable estimation. We new explanation into the discussion section.

*Section 5* I assume that the simulations presented in this section have been run on some Nvidia GPU card since the authors previously mentioned some CUDA files. However, this should be stated again here, as well as mentioning what exact GPU card was used and how many of them were needed to run the high resolution models.

This is a good point and we added this information.

\subsection{Implementation using Graphical Processing Units (GPUs)}

The initial code prototyping was conducted on a laptop equipped with a 13th Gen Intel Core i9-13900HX CPU (64GB RAM) and an NVIDIA GeForce RTX 4090 (16 GB) laptop GPU. For large-scale 3D simulations, the computations were carried out on an NVIDIA DGX-1-like node, featuring 4 NVIDIA Ampere A100 GPUs (each with 80 GB of memory) and an AMD EPYC 7742 server processor with 512 GB of RAM.

*Section 5.1* Before jumping into eq. 65, I believe it's a good idea to briefly introduce the plastic model of Duretz et al 2019, perhaps even adding a small sketch with the elastic springs, dampers and whatnot. This would also help readers

unfamiliar with this plastic model understand why theres a viscous damper in the yield function.

We agree with the reviewer that some explanation might be needed. That's why we refer to Duretz et al 2019. We added more references for an interested reader.

Resolving strain localization in frictional and time-dependent plasticity: Two- and three-dimensional numerical modeling study using graphical processing units …
Y Alkhimenkov, L Khakimova, I Utkin, Y Podladchikov

An interested reader may refer to \cite{alkhimenkov2024shear, https://doi.org/10.1029/2023JB028566} for more details on the implementation of plasticity.

Shear bands triggered by solitary porosity waves in deforming fluid-saturated porous media
Y Alkhimenkov, L Khakimova, Y Podladchikov
Geophysical Research Letters

The constants A, B, C are merely some trigonometric functions. I don't think there is any need of re-binding them with new names; they only appear in two equations, and since these equations are usually well-known for a wide spectrum of the potential readers, the new names just make the equations more confusing.

We agree with the reviewer that some explanation is needed. There are different definitions of A, B and C in plasticity and we used only a particular one. Keeping the same notation in Eq 65-66 make these equations more universal.

*L385* Perhaps not every reader know under what conditions a material is within the plastic regime. It would be helpful to add that this happens when $F^{("trial")} > 0$

We agree with the reviewer that some explanation might be needed. This study is about APT methos and not about plasticity. That's why we refer to Duretz et al 2019. We added more references for an interested reader and a sentence with explanation to fulfill the reviewer request.

An interested reader may refer to \cite{alkhimenkov2024shear, https://doi.org/10.1029/2023JB028566} for more details on the implementation of plasticity.

Shear bands triggered by solitary porosity waves in deforming fluid-saturated porous media
Y Alkhimenkov, L Khakimova, Y Podladchikov
Geophysical Research Letters

*Section 5.2* I assume the domain of the model is $\Omega \in [0,1] \times [0,1]$; however, this should be explicitly stated in the text.

This is a good point and we added this information.

Let us consider a 2D numerical domain with $L_x=L_y=1$.

Is a resolution of $10000^2$ really necessary? Did the authors run systematic tests to explore whether one can get a way with lower resolutions?

Yes, the resolution $10,000^2$ is necessary to show the robustness of the APT method. We add a reference where systematic tests were performed with different resolutions.

Resolving strain localization in frictional and time-dependent plasticity: Two-and three-dimensional numerical modeling study using graphical processing units (GPUs). Y Alkhimenkov, L Khakimova, I Utkin, Y Podladchikov

How does the convergence of this highly-nonlinear setup behave? Is every single time step fully converged? Would be interesting to plot also (number of iterations / nx) vs time step, I suspect the number of PT iterations increases when plasticity kicks in. How much time does it take to run a model with this resolution? Same comments apply to Section 5.3

Yes, every iteration converged. The plot requested by the reviewer already exists in Figure D1 (in the present simulation is similar). In

Resolving strain localization in frictional and time-dependent plasticity: Two-and three-dimensional numerical modeling study using graphical processing units (GPUs). Y Alkhimenkov, L Khakimova, I Utkin, Y Podladchikov

The present convergence is fully analogous to Fig D1 in the article above. Adding such a technical detail (plot) is not possible because it will require re-running the HR simulation.

The simulation time takes about a few hours.

*Figure 7* Put the spatial coordinates in the labels of the x and y axes instead of the grid cell numbers. Also, this figure alone does not bring much, it could probably be merged as a fourth panel in Fig 8.

We put cells numbers in Fig 7,8,9 to show the resolutions employed, this was done on purpose. We have separate Fig 7 and 8 because we would like to have full size of Fig 8 to show fine details of the strain localization.

*L400* It would be nice if the authors could add a few more snapshots of models at much lower resolution to make stronger the argument that the strain localisation is mesh-independent.

We refer to our resent study, where more models were investigated using the same regularization method:

Resolving strain localization in frictional and time-dependent plasticity: Two-and three-dimensional numerical modeling study using graphical processing units (GPUs). Y Alkhimenkov, L Khakimova, I Utkin, Y Podladchikov

*Figure 8* I may be wrong, but the colour scale of panel B seems to have slightly different min/max values with respect to panels A and C

Yes, there is a slight difference, we made it on purpose to better visualization. As long as the color bar attached – any scales should be accepted.

*Figure 9* As Fig 7, it could be merged with Fig. 10

We combined Fig 9 and 10 as suggested by the reviewer.

*Section 5.3* I am not so sure I would call this "ultra-high" resolution. This resolution fits without many problems in a single modern GPU card, and given that only 15 time steps are performed, it should run in just a few hours if it converges fast enough.

Yes, the reviewer is correct, it fits into a single GPU card that has 80 GB of DRAM memory. The term Ultra-high is chosen because as far as we know, there are no simulations with such a resolution yet in the literature.

*Section 6* One could add here a brief intro of this section.

We added some introduction into this section.

In this section, we analyze the implications of the numerical results presented in the previous sections and establish connections with relevant works in the field. We explore the behavior of the numerical parameters, such as the Strouhal number ($\mathrm{St}$), and their optimal values for different physical models including elastic, viscoelastic, and poroelastic media. Additionally, we assess the influence of dimensionality, initial and boundary conditions, and non-linearities such as plasticity on the convergence and accuracy of the simulations. This analysis serves as a foundation for further extending these methods to more complex and realistic scenarios.

*Section 6.3* It is not very clear whether these simulations were run for the paper here referenced, or they are some other simulations not described in this manuscript. If these are simulations from a previous paper, why not use the ones here presented? If they are actually new simulations, please describe these models in detail.

We performed these simulation in section 6.3 only for the present study. The purpose of this section it to show how St differs with respect to boundary conditions. We rely on the recent paper by Rass et al 2022, where no detailed explanation of all simulations is performed. Since it will not add something new into the article, we keep the present brevity as Rass et

al 2022. However, we added some more explanations for reproducibility. Also note that full results can be reproduced since all the codes are shared via zenodo.

Räss, L., Utkin, I., Duretz, T., Omlin, S., and Podladchikov, Y. Y.: Assessing the robustness and scalability of the accelerated pseudo-transient method, Geoscientific Model Development, 15, 5757–5786, 2022.

To what time step (or point in the stress-strain curve) do these plots correspond to? is plasticity kicking in already from the first time step? How do these plots vary for simulations along different points of the stress-strain curve?

First several time steps are purely elastic. In the middle of the simulation, stress reaches yield and the plots we are showing in the discussion section correspond to the condition when plastic flow is activated.

These simulations correspond to the loading scale where plastic flow is activated.

*L511* I don't think this is a conclusion related to the work here presented.

To reflect this comment we improved the conclusion, made it more concise and more related to the article.

Albert de Montserrat

We thank the reviewer for the in-depth comments and for correcting typos, which helped us improve the quality of the manuscript.

Sincerely,
Yury Alkhimenkov and Yury Podladchikov